# How Will I Evaluate Others? The Influence of “Versailles Literature” Language Style on Social Media on Consumer Attitudes Towards Evaluating Green Consumption Behavior

**DOI:** 10.3390/bs15070968

**Published:** 2025-07-17

**Authors:** Huilong Zhang, Huiming Liu, Yudong Zhang, Hui He

**Affiliations:** School of Economics and Management, Jiangxi Normal University, Yaohu Campus, No. 99, Ziyang Avenue, High-Tech Development Zone, Nanchang 330022, China; zhlrex@foxmail.com (H.Z.); 202340100655@jxnu.edu.cn (H.L.); 202440100888@jxnu.edu.cn (H.H.)

**Keywords:** language style, green consumption behavior, social media, Versailles Literature

## Abstract

The dissemination and practice of green consumption behavior is an important issue in promoting sustainable development. With the advent of the digital age, social media platforms have become an important channel for promoting green consumption. The expression of language style has become an increasingly important factor influencing consumer attitudes. From the perspective of consumer perception, this study used three situational simulation experiments (*n* total = 304) to explore the mechanism by which the “Versailles Literature” language style impacts the feelings and behaviors of audiences of the green consumption behavior of the poster, and to examine the mediating roles of ostentation perception and hypocrisy perception. Data analysis was conducted using SPSS. The research findings showed that, compared with “non-Versailles Literature”, this style significantly reduces positive attitudes toward green consumption while increasing perceptions of bragging and hypocrisy. Furthermore, the strength of social ties between the consumer and the poster plays a moderating role in the effect of language style; specifically, when posts come from strangers, consumers perceive a stronger sense of bragging and hypocrisy. The research results will provide practical guidance for individuals and enterprises to effectively promote the concept of green consumption on social media, helping enterprises avoid the negative reactions brought about by conspicuous green consumption behaviors and exaggerated or false promotion of environmental behaviors, such as “greenwashing”.

## 1. Introduction

In the age of digitalization, social media has become an essential medium for information dissemination and interaction. Under global challenges such as energy shortages and environmental pollution, as well as the promotion of China’s “dual carbon” policy and the advocacy for green consumption policies, social media platforms play a significant role in promoting green consumption behaviors ([1]). Unlike traditional communication channels, social media is characterized by user-generated content (UGC), which allows individuals to share green consumption practices in more personalized, socialized, and visualized ways. By actively encouraging users to engage with environmental topics and facilitating the sharing of user-generated content—such as posts about eco-friendly actions or green product recommendations—social media significantly contributes to the dissemination and adoption of sustainable green consumption culture ([76]). However, in the cultural context of China, the face culture and collectivist values make the language style adopted by users when displaying green consumption behavior an important factor influencing others’ perception and evaluation. For example, some influencers highlight their environmental efforts through plain and simple expressions, receiving the support and emulation of followers, thus driving the trend of sustainable green consumption. However, in the internet age, where distance and subjectivity are diminished in the online society, some users exaggerate their green consumption behaviors through “Versailles-style” expressions; that is, through the language style of seemingly low-key expression to imply their achievements. This aims to showcase their environmental achievements and construct an image of themselves as “eco-pioneers”, such as “It is really tiring to do environmental protection, but in order to protect the earth, I am willing to stick to it.” Furthermore, some enterprises engage in “Greenwashing” through superficial environmental protection publicity; that is, enterprises exaggerate or falsely promote their environmental protection behaviors. This undermines consumer trust in green marketing, increases public skepticism and resentment, and impedes the spread of green consumption behaviors. This presents a critical question: how do different linguistic styles on social media affect consumers’ attitudes toward the green consumption behaviors of posters, as well as the practice of sustainable green consumption and the dissemination of green information?

In recent years, scholars have extensively explored the role of social media in promoting green consumption, with a particular focus on how user-generated content, social network relationships, and Eco-labels influence consumers’ willingness to engage in green behaviors ([7]; [59]). These studies emphasize the positive role of social media in fostering green consumption behaviors, particularly through group-based dissemination and interactive effects. Existing research has primarily focused on green advertising, travel reviews, influencer word-of-mouth, and brand attitudes. Scholars argue that the manner of linguistic expression significantly influences consumers’ attitudes toward brands or products ([13]; [60]; [24]; [81]). For example, scholars have found that the use of bragging language styles in advertisements can lead consumers to have a negative attitude towards the brands ([67]). However, green marketing research focuses mainly on corporate communication strategies (such as green advertising and eco labels) and commercial communication contexts while neglecting the influence of individual consumer language expression styles on green consumption attitudes and in-depth exploration of personal sharing scenarios on social media. In particular, although the emerging “Versailles Literature” language style has become popular in online culture, its psychological mechanisms and influence in the context of green consumption have not been systematically explored.

Based on self-presentation theory, this study uses an experimental research method from the perspective of consumer perception to explore the differences in audience’s perception of green consumption behavior attitudes when users share such behaviors using different language styles in a social media context. Theoretically, this research will enrich our understanding of the relationship between emerging language expression styles, green consumption behavior, and social media communication, particularly the role of “Versailles Literature” language style in green consumption communication. It also offers a new perspective on social media communication and consumer behavior. On a practical level, this study offers valuable guidance for optimizing green consumption communication strategies on social media for both individuals and enterprises. It helps enhance the effectiveness of language style in communication while avoiding negative perceptions caused by boastful expressions or “greenwashing”. The findings contribute to encouraging public acceptance and practice of sustainable green consumption and supporting the achievement of sustainable development goals.

The structure of this paper is as follows: Section 2 reviews relevant theories and prior research; Section 3 presents the research model and hypotheses; Section 4 describes the experimental design and data analysis; Section 5 discusses the research findings and practical implications; Section 6 notes the limitations of this study and future work.

## 2. Theoretical Background

### 2.1. Language Style and Versailles Literature

Language style refers to the way in which people convey information during communication; that is, the content of social significance is conveyed in this way ([16]). Language style can demonstrate an individual’s unique identity and attitude through vocabulary, sentence structure, and intonation. It also influences how information recipients cognitively and emotionally respond to messages. Individuals convey not only information content through language expression, but also signals of their intrinsic values and moral character. According to the theory of moral inference, the audience will automatically make moral evaluations and motivational attributions based on the speaker’s language style, which will directly affect their judgment of the authenticity of the speaker’s behavior ([17]; [31]). Existing studies have categorized language styles into abstract and concrete, metaphorical and literal, and rhetorical and declarative, etc. ([12]; [56]; [85]), highlighting their significant role in information dissemination.

With the popularity of social media, the ways through which online users express themselves are beginning to show diverse and individualized characteristics. One emerging language style that has attracted the attention of researchers is the “Versailles Literature” language style. The term “Versailles Literature” originates from the Japanese manga The Rose of Versailles, which vividly depicts the luxurious and extravagant life of the French nobility at the Palace of Versailles in the 18th century. Chinese netizens used the term “Versailles” as a metaphor to describe a narrative style that subtly showcases a superior lifestyle through linguistic finesse. This concept closely aligns with the Western academic notion of “humblebragging”, which is defined as a covert form of self-promotion disguised through surface-level strategies such as complaining, questioning, self-deprecation, narration, or humor ([78]; [86]). Scholars have defined “Versailles Literature” from various perspectives. From a linguistic perspective, it is considered a distinctive rhetorical device that employs exaggeration, irony, and metaphor to achieve a dual effect of flaunting and self-deprecation ([54]). Psychologically, it is viewed as a form of self-presentation and identity construction, reflecting individuals’ psychological needs and behavioral motivations on social media platforms ([36]). Sociologically, “Versailles Literature” is seen as a symbolic expression of social identity and status, revealing underlying identity anxiety and status competition in modern society ([50]).

Therefore, in the context of green consumption, this study defines the “Versailles Literature” language style as a rhetorical approach that employs irony, exaggeration, and contrast to subtly showcase one’s achievements in green consumption under the guise of modesty, complaint, or self-deprecation. Its main features include a structure of initial understatement followed by implicit praise, reverse comparison, contextual dependency, and often the use of a third-person narrative perspective ([62]; [34]). However, as a special form of expression, the “Versailles Literature” language style’s implicit self-glorification and display of superiority go against the social moral expectations of honest communication and humility, thus possessing unique moral hazard characteristics ([58]). This differs from traditional forms of “direct bragging” and “virtue signaling”. The former refers to overt and unapologetic self-praise to highlight personal achievements, while the latter focuses on displaying moral superiority, such as emphasizing eco-friendly behaviors to signal ethical values ([29]; [43]). In contrast, the “Versailles Literature” language style aims to guide audiences toward recognizing the speaker’s material advantages and accomplishments through seemingly humble or complaining statements that actually serve as subtle boasts (See Table 1 for details). Its covert and conspicuous nature makes it difficult for the audience to directly identify its true intention, and it is more likely to raise questions about the speaker’s motivation. The increase in cognitive load leads to the audience’s perception of hypocrisy and even negative emotions ([10]).

In recent years, scholars have conducted extensive research on the application and effects of language styles on social media. The existing studies mainly focus on individuals’ self-presentation, such as boasting, humility, or humor, as well as the effects of information dissemination and attitudes of receivers ([47]; [60]). For example, [72] ([72]) pointed out that language style can reflect the speaker’s social status and power relationship, while [5] ([5]) further emphasized the role of language style in self-presentation on social media, suggesting that different language styles may influence others’ interpretations of the speaker’s motivation.

The theoretical foundation of the “Versailles Literature” language style can be traced back to [26]’s ([26]) theory of “self-presentation strategies”. Goffman argued that individuals use specific linguistic and behavioral tactics in social interactions to shape their impressions. As a unique form of self-presentation, the “Versailles Literature” language style has drawn scholarly attention to various factors that influence its communicative effectiveness on social media. These include cultural background, gender, and the perceived expertise of the influencer, all of which can shape how “Versailles Literature” language style messages are received and interpreted in digital environments ([38]; [21]; [81]). For example, cultural differences can lead audiences to interpret “Versailles Literature” language style expressions in varied ways; the perceived expertise of an influencer can significantly impact the credibility and persuasiveness of such communication. Furthermore, existing research has explored the positive effects of the “Versailles Literature” language style on brand attitudes and purchase decisions of consumers. When influencers promote luxury brands on social media using this style, it can evoke feelings of admiration and benign envy among consumers, thereby enhancing their favorability toward the brand, improving attitudes, and increasing purchase intentions ([48]; [54]). [40] ([40]) pointed out that the Japanese snack brand Pocky widely employs rhetorical devices such as exaggeration, metaphor, personification, and conjunction omission in its promotional materials. Notably, in its 2022 campaigns, the brand utilized “Versailles Literature” language to create a dramatic product impression, effectively achieving its branding objectives through a distinctive language style. The “Versailles Literature” language style exhibits a complex mechanism of influence in the context of green consumption behavior. Within this domain, it can serve as a strategic tool for brands to build a positive image of sustainability and to stand out in a highly competitive market ([61]). However, the use of “Versailles Literature” language style in green consumption communication may also lead to negative effects. Scholars suggest that an influencer’s use of the “Versailles Literature” language style to boast about their green consumption behaviors can influence consumers’ perceptions of the brand’s altruism, leading to negative attitudes toward both the influencer and the promoted brand ([10]).

Although existing studies have explored the role of the “Versailles Literature” language style in self-presentation, branding, and luxury marketing, its application in green consumption and the underlying mechanisms affecting consumer attitudes remain under-researched. In particular, little attention has been paid to how the “Versailles Literature” language style influences consumers’ interpretation of the poster’s motives in the context of green consumption, thereby shaping their attitudes toward and acceptance of others’ green behaviors. The boundary conditions under which this language style operates effectively in green consumption contexts also need further investigation.

### 2.2. Hypocrisy Perception

Hypocrisy perception refers to an individual’s subjective judgment about the inconsistency between others’ words and actions or the lack of authenticity in their motivations ([75]). In the context of green consumption, this study defines hypocrisy perception as consumers’ skepticism about the authenticity of the promoter’s environmentally friendly behaviors. Hypocrisy perception focuses more on the authenticity and consistency of the behavior of the sender, such as the belief that companies use environmental behavior to attract public attention rather than being truly committed to sustainable development. When a company’s actual environmental performance falls short of the expectations established by its propaganda, consumers develop cognitive dissonance, resulting in a perception of hypocrisy ([23]). Its essence is the psychological reaction of consumers when they perceive that there is a significant gap between environmental publicity and actual behavior. Hypocrisy perception has become an important topic in consumer behavior research in recent years, particularly in fields such as ethics, green consumption, and social media communication. In the context of the increasing popularity of green marketing, the authenticity of environmental protection publicity in enterprises has become increasingly prominent, and the phenomenon of greenwashing has become a concrete manifestation in the field of green marketing. Greenwashing refers to the behavior in which enterprises use false, exaggerated, or misleading environmental statements to shape their image of environmental protection, but in fact do not really implement the corresponding environmental protection measures. When companies exaggerate or falsely publicize their environmental behavior, but fail to fulfill their environmental commitments, consumers perceive that the company’s environmental publicity and actual behavior are inconsistent, resulting in a strong sense of hypocrisy ([66]; [19]). In the social media context, hypocrisy perception can lead consumers to question the motives of the sender and even lead to widespread negative communication ([3]). Research shows that hypocrisy perception stimulates negative emotions such as anger, disappointment, and distrust, which lead to deterioration of consumer attitudes and boycott behavior ([77]). Therefore, when consumers perceive hypocrisy in the promotion of green behaviors by companies or individuals, it significantly reduces their trust in them, thereby impacting their willingness to purchase green products ([35]).

Although hypocrisy perception has received widespread attention in the fields of moral behavior, green consumption, and social media communication, current research mainly focuses on the actions of brands and companies, with relatively little exploration of individual users’ behavior presentations on social media. Specifically, in the context of green consumption behaviors, there is a need for further investigation into how hypocrisy perception under emerging language styles (such as Versailles Literature) affects the audience’s emotional reactions (e.g., anger, disappointment) and specific behaviors (e.g., negative comments, refusal to emulate). This gap in the research calls for deeper exploration into the impact of language styles on consumer behavior and attitudes toward green consumption.

### 2.3. Bragging Perception

Bragging is a self-presentation strategy that typically arises from an individual’s psychological state caused by the failure to fully satisfy their desire for respect ([55]). Bragging perception refers to an individual’s subjective perception of others showcasing personal achievements or status through specific behaviors, language, or symbols. At its core, it is the observer’s interpretation of the motives and appearances of the information transmitter, usually linked to “boasting” or “self-enhancement” ([6]). Existing research primarily uses social comparison theory to explain bragging perception. According to social comparison theory, when audiences observe this self-presentation strategy, they often spontaneously compare themselves, and if they perceive themselves as being in a disadvantaged position, they are more likely to perceive others’ behaviors as ostentatious ([45]). The formation of bragging perception is often associated with conspicuous consumption, such as individuals attracting attention by displaying high-end brands, sharing luxurious trips, or subtly boasting about personal achievements. These behaviors are typically interpreted by observers as being motivated by bragging. The core logic of ostentatious consumption has continued to develop in the context of green consumption, accompanied by corresponding emotional and behavioral reactions. In green consumption contexts, individuals often display their sense of social responsibility and superiority through the purchase or display of environmentally friendly products or services. The bragging perception can trigger negative emotions, such as jealousy or resentment, and weaken trust and identification with the individual, potentially even leading to resistance behaviors ([44]).

## 3. Research Hypotheses

According to [26]’s ([26]) self-presentation theory, individuals often adopt specific linguistic and behavioral strategies during social interactions to shape how they are perceived by others and gain the respect of others. The classical two-factor model proposed by [46] ([46]) shows that impression management is composed of two core components: impression motivation and impression construction. Studies by [69] ([69]) and [4] ([4]) found that “Versailles Literature” has a dual role. Individuals highlight their strengths and achievements through conspicuous behavior, which aims to build cognitive competence and win the respect and recognition of others. Individuals can also use strategies such as humble expression and moderate complaint to create a prosocial image, which can not only stimulate others’ sympathetic response, but also convey intimacy signals and trust hints, thus promoting the establishment of loving relationships. This reflects that two core views of individual self-presentation are highly similar: One is to seek liking motivation; that is, to obtain the emotional identity of others by shaping an attractive and affinity self-image ([33]). The second is to respect acquisition motivation, which is to build authoritative awareness and win the respect of others by demonstrating personal abilities, achievements, and values ([9]; [54]). Therefore, “Versailles Literature” is a special expression phenomenon in the era of social media, and is often used to seek self-improvement and win the recognition and love of others. People often use “Versailles Literature” language style on social media to boost their self-image and earn others’ approval and admiration. “Versailles Literature” language style often attempts to mask its underlying boastful intent through modesty and complaint, which leads consumers to perceive it as a disingenuous form of self-presentation. This perceived insincerity can trigger negative emotions toward the poster, reducing both likability and trust. For instance, when CEOs engage in self-praise using “Versailles Literature” language style, investors may form negative impressions and become less willing to invest ([29]). Similarly, [51] ([51]) found that compared with straightforward boasting, using a “Versailles Literature” language style approach to self-promotion on Facebook diminishes perceived sincerity and likability.

Consumers may believe that a poster deliberately uses green consumption behaviors to cultivate a noble social image. When consumers perceive that the behavior is driven by social status or self-promotion rather than genuine environmental motivation, perceived hypocrisy increases ([30]). Studies indicate that when customers perceive hypocrisy in a company’s corporate social responsibility (CSR) activities, they tend to generalize this perception. This leads to negative overall evaluations of the company and its brand, which further reduces their purchase intention and related behaviors ([2]). Negative consumer reactions, such as word-of-mouth, complaints, and resistance behaviors, might result from a company’s irresponsible environmental actions ([79]). To gain attention or admiration, posters who use Versailles Literature to deliberately exaggerate or accentuate their green consumption behaviors may reduce the perceived credibility of the information source, enhancing the perception of hypocrisy, leading to a negative attitude toward the poster and the recommended brand.

When individuals exaggerate their achievements or flaunt their comparative advantages, audiences may experience envy triggered by social comparison, especially when their own needs are perceived as unmet ([74]). Studies have shown that when product users in advertisements employ “Versailles Literature” language style approach to express superiority—masking boastful motives behind a facade of modesty or complaint—audiences often interpret this as insincere. This perceived insincerity can undermine the audience’s self-esteem and self-worth, ultimately leading to negative evaluations of the advertisement ([83]). Therefore, when a post uses the “Versailles Literature” language style to describe its green consumption behavior, the social comparison mechanism urges consumers to compare the post’s expression with their own or others’ environmental behavior. Consumers may think they are acting not to protect the environment itself, but to show off their economic ability or social status. When individuals perceive other people’s ostentatious behavior, they will feel threatened and respond defensively, believing that the ostentatious try to improve their status by belittling others ([41]). Therefore, in the context of green consumption, the implicit flaunting characteristics of the “Versailles Literature” language style make consumers believe that the sender is not motivated by real environmental protection, but is using green consumption as a tool to show their moral superiority and economic strength. This leads to consumer disgust and eventual negative attitudes towards their green consumption behavior. In contrast, the “non-Versailles Literature” language style is more likely to be seen as direct and humble, thus more likely to be perceived as sincere behavior. Therefore, when consumers perceive that a poster is deliberately creating a noble, environmentally friendly social image rather than authentically reflecting their internal values, the consumers’ perceptions of hypocrisy and bragging will be amplified, leading to negative attitudes toward the poster’s green consumption behaviors and even evoking feelings of aversion. Based on this, we propose the following hypotheses:

**H1:** *The “Versailles Literature” language style is more likely to trigger negative consumer attitudes towards the green consumption behavior of the sender than a non-Versailles language style*.

**H2a:** 
*Hypocrisy perception plays a mediating role in the mechanism through which “Versailles Literature” language style affects consumers’ negative attitudes towards green consumption behaviors.*


**H2b:** 
*Bragging perception plays a mediating role in the mechanism through which “Versailles Literature” language style affects consumers’ negative attitudes towards posters’ green consumption behaviors.*


The strength of social ties refers to the closeness of social connections, primarily reflected in the intimacy between individuals and the emotional intensity invested in social interactions ([11]; [28]). [73] ([73]) pointed out that psychological distance affects individual thinking constructs; closer psychological distance causes people to adopt low-level and specific constructs, while longer psychological distance leads to high-level and abstract constructs. Individuals with strong ties share a high degree of intimacy, frequent interactions, and enduring emotional connections, which makes them more attuned to emotional value ([14]; [25]). This often results in shared social identity and a sense of belonging, and the behavior of members of the group is more likely to be positively evaluated and understood. Therefore, when there is a strong relationship connection, consumers will regard the sender as a member of the group and even in the face of language expression with conspicuous characteristics, they will be more inclined to interpret it as a normal way of communication or personal expression style within the group ([71]). Therefore, due to the close social distance and deep trust foundation in a strong relationship connection, consumers tend to use low-level constructs for information processing, focusing more on specific contextual factors than on abstract characteristics. Consumers will think that it is a situational need rather than intrinsic hypocrisy motives, and thus will extend more goodwill, understanding, and tolerance to their green consumption behavior. When a review is written by a source with strong ties to the consumer, the consumer views the author as a credible information source ([49]; [52]). When the information source has a close relationship with the consumer, the consumer is more likely to trust the source, believing in their advice and making corresponding behavioral decisions based on it. When a poster uses the “Versailles Literature” language style, consumers tend to interpret the poster’s actions from a benevolent perspective, assuming that the green consumption behaviors are more likely driven by genuine environmental motives. In close relationships, individuals often feel responsible for maintaining social connections within their networks ([63]). When consumers perceive that a poster is boasting about their environmental achievements, they may worry that overly criticizing the poster could negatively affect emotional connections within their social network; as a result, they may be more inclined to reduce negative evaluations of the poster. Drawing on previous research, we propose that the strength of social ties moderates the influence of consumers’ attitudes toward the poster’s green consumption behaviors.

In contrast, weak ties are characterized by lower interpersonal intimacy and emotional intensity, where individuals are more likely to focus on informational value ([27]; [82]). According to the self–other perspective difference theory ([37]), in weak ties, the level of intimacy and emotional intensity between individuals is lower, and consumers often rely on surface cues (such as the “Versailles Literature” language style) to infer the poster’s motivations. Due to the weaker trust foundation in weak ties, consumers tend to assess the poster’s intentions based on overt behaviors, such as language style. Therefore, due to the lack of deep trust foundation and social distance in the weak relationship connection, consumers use more high-level constructs, rely on surface clues such as language style to infer the real motivation of the sender, and regard the implicit flaunting characteristics in “Versailles Literature” language style as evidence of hypocritical environmental protection or greenwashing behavior ([64]; [57]). In this context, the subtle boasting characteristics in the “Versailles Literature” language style lead consumers to believe that the poster is exaggerating their environmental achievements to gain recognition, rather than having genuine environmental intentions. As a result, consumers interpret the behavior as hypocritical green consumption or greenwashing, which triggers a stronger sense of hypocrisy and bragging, a negative attitude towards the green consumption behavior of the sender, and may trigger a stronger aversion. Therefore, the strength of social ties moderates consumers’ attitudes toward the poster’s green consumption behaviors to some extent. Consumers in strong ties are more likely to assign higher credibility and interpret the poster’s actions with goodwill, reducing the negative impact of bragging perception and hypocrisy perception. In contrast, consumers in weak ties are more likely to infer motivations from surface characteristics, intensifying their negative attitudes toward the poster. Based on this, we propose the following hypotheses:

**H3:** 
*The strength of social ties plays a moderating role in the mechanism of “Versailles Literature” language style affecting consumers’ attitudes towards green consumption behavior.*


**H3a:** 
*Strong social ties moderate the effect of the “Versailles Literature” language style on consumers’ perceptions of hypocrisy, thereby reducing negative attitudes toward the poster’s green consumption behaviors.*


**H3b:** 
*Strong social ties moderate the effect of the “Versailles Literature” language style on consumers’ perceptions of bragging, thereby reducing negative attitudes toward the poster’s green consumption behaviors.*


In summary, the conceptual model is illustrated in Figure 1.

## 4. Experimental Design and Testing

### 4.1. Overview of Studies

The purpose of this study was to explore the impact of Versailles literature on green consumption behavior and its mechanism. The main effect, mediating effect, and moderating effect of the “Versailles Literature” language style on green consumption behavior were verified using three experiments: In Experiment 1, a one-way intergroup design (Language Style: Versailles Literature vs. non-Versailles Literature) was used to verify the direct impact of “Versailles Literature” language style on green consumption behavior attitude and to explore its mediating mechanism. Study 1 showed that “Versailles Literature” language style is more likely to trigger negative consumer attitudes towards green consumption behavior than “non-Versailles Literature” language style. Bragging perception and hypocrisy perception play a mediating role in this mechanism. Experiment 2 further introduced the strength of social ties as a moderating variable and used a two-factor design of 2 (language style) × 2 (strength of social ties: stranger vs. acquaintance) to investigate the moderating effect of strength of social ties on “Versailles Literature” language style on green consumption behavior. Experiment 3 strengthened the moderating effect of the strength of social ties through a platform-differentiated design and further enhanced the moderating effect of the strength of social ties (See Table 2 for details). Studies 2 and 3 showed that the strength of social ties plays an important moderating role in this process; for example, when consumers see “Versailles literature” posts published by strangers, they perceive a stronger sense of bragging and hypocrisy, leading to negative attitudes towards the green consumption behavior of posters.

### 4.2. Experiment 1

#### 4.2.1. Pilot Experiment

The purpose of the preliminary experiment was to manipulate and test the content of the “Versailles Literature” and “non-Versailles Literature” language style posts used in the formal experiment. In terms of textual information, this study designed two pieces of content to ensure consistency in word count, font, and information volume, avoiding any interference from visual factors on cognition. The “Versailles Literature” language style post content was “I originally thought that planting trees was as simple as donating a few. After all, people like me have limited time, and who has the time to worry about these things? But then I thought, donating trees matches my exceptional temperament and sense of social responsibility. Since reducing carbon emissions seems to depend on me, I reluctantly decided to donate 50 trees, which I only spent 500 yuan.” The “non-Versailles Literature” language style post content was “Recently, I joined the Green Future Carbon Neutrality Project and donated 50 trees, costing 500 yuan. I believe environmental protection is not only a personal responsibility but also an action that aligns with my values. By reducing over a thousand pounds of carbon emissions and participating in the tree seedling exchange program through waste recycling points, it is a fulfillment of social responsibility and a contribution to the future. It makes me feel very meaningful.”

Next, participants were asked to rate the “Versailles Literature” language style on a scale by responding to the question: “Do you think this text represents a “Versailles Literature” language style? (“Versailles Literature” language style refers to the use of apparent modesty, complaint, or self-deprecation to subtly convey one’s superior status or achievements).” All related variables were measured using a 5-point Likert scale, where 1 indicates “strongly disagree” and 5 indicates “strongly agree.” The preliminary experiment was conducted using a questionnaire platform, collecting 30 valid responses (53.33% male, aged 18–30). The results of the *t*-test showed that the difference in scores between the “Versailles Literature” language style (M_Versailles Literature = 4.1, SD = 0.6618) and the “non-Versailles Literature” language style (M_non-Versailles Literature = 1.733, SD = 0.5833) posts was significant (t(29) = −14.695, *p* = 0.008). Therefore, these two pieces of content were selected for the formal experiment.

#### 4.2.2. Formal Experiment

The purpose of Experiment 1 was to verify the primary impact of “Versailles Literature” language style and “non-Versailles Literature” language style on consumers’ attitudes toward the poster’s green consumption behaviors, as well as the mediating effects of bragging perception and hypocrisy perception. Experiment 1 employed a between-subjects design with a single factor (Language Style: Versailles Literature vs. Non-Versailles Literature). The dependent variable was consumers’ attitudes toward the poster’s green consumption behaviors. A total of 60 students from a university were randomly divided into two groups for the experiment, with participants randomly assigned to one of the two groups. Before recruiting participants, we used G* power 3.1 software for efficacy analysis to ensure a statistically sufficient sample size. Based on the recommendations of [15] ([15]), the equivalent stress (d = 0.80), the significance level alpha = 0.05, and the statistical efficacy 1-beta = 0.85 were set, and the minimum number of participants in each group was calculated to be 30, with a total sample size of 60. In this study, we chose college students as the research object, considering the characteristics of college students in terms of cultural background, socioeconomic status, and age structure. College students often have strong social media skills and are highly sensitive to popular online culture, making them an ideal group in which to study digital interactions and the impact of social media. This study issued a recruitment announcement through the Student Affairs Office of a university and invited college students to participate in the experiment via random sampling. The recruitment announcement detailed the research purpose, participation requirements, and reward measures for participants. A total of 60 students were recruited and divided into two groups to participate in the experiment. All participants were students who had not been exposed to the experiment, and the participants did not receive relevant training or pre-experimental materials before the experiment. The 60 participants were randomly assigned to the two experimental groups using the random number generation function in the SPSS software. All participants signed informed consent forms, and this study was reviewed by the university’s ethics committee (IRB-jxnu-b-20241101, 1 November 2024) and conducted in accordance with the Helsinki Declaration. No personal identification information was collected.

First, participants were instructed to browse a post shared by someone on social media, detailing their recent green consumption behavior experience. The stimulus material used in Experiment 1 is the same as the pilot experiment material. After reading the post, participants were asked to answer several questions to assess their perceptions of bragging and hypocrisy, as well as their attitudes toward the poster’s green consumption behavior. They were also asked to provide their demographic information, such as age and gender. The bragging perception scale was adapted from [65] ([65]), with items such as “I think the poster is showing off their green consumption behavior (BP1),” “The poster is trying to demonstrate their superiority through this (BP2),” and “This post gives me a sense of bragging (BP3).” The hypocrisy perception scale was adapted from [22] ([22]) to evaluate items such as “I think the poster’s behavior is hypocritical (HP1),” “I believe the poster is not genuinely engaged in environmental protection (HP2),” and “The poster’s behavior makes me feel it is for show, not for environmental protection(HP3).” The green consumption behavior attitude scale was adapted from [20] ([20]), with items such as “I believe the poster’s behavior has a positive impact on the environment (LS1),” “I admire the poster’s green consumption behavior (LS2),” and “I have a positive attitude toward this poster’s green consumption behavior (LS3).” To minimize potential confounding effects from consumers’ awareness of green consumption behavior and face-consciousness, participants also reported on these variables, such as green consumption behavior awareness (e.g., “I tend to choose environmentally friendly products”) and face-consciousness (e.g., “I tend to choose products or services that enhance my image and display my environmental behavior in public”). By measuring these variables, potential confounding factors could be effectively controlled, allowing for a more accurate assessment of the influence of “Versailles Literature” and “non-Versailles Literature” language styles on consumers’ attitudes toward the poster’s green consumption behaviors. All measurements of the relevant variables were conducted using a 5-point Likert scale, with responses ranging from 1 (strongly disagree) to 5 (strongly agree).

#### 4.2.3. Experimental Results

Firstly, the manipulation check results show that participants perceived significant differences in the language styles. Specifically, the measurement values for the “Versailles Literature” language style post (M_Versailles Literature = 3.93, SD_Versailles Literature = 0.907) were significantly higher than those for the non-Versailles Literature language style post (M_non-Versailles Literature = 2.17, SD_non-Versailles Literature = 0.834, *p* < 0.001), confirming the successful manipulation of the language style variable.

Basic participant information, including age, gender, and occupation, is summarized in Table 3. The reliability of the scale was assessed using SPSS 23.0 software, and AMOS 24.0 was employed for confirmatory factor analysis. Table 4 contains the standardized factor loading coefficients, AVE, CR, and Cronbach’s α values for the variables examined in this study. The convergent validity of bragging perception, hypocrisy perception, and consumers’ attitudes toward green consumption behavior is well established, as all values exceed the threshold of 0.5. The square root of the AVE of each variable is greater than the correlation coefficient between this variable and other variables, and the HTMT value between all variables is lower than the threshold of 0.90, indicating that the scale in this study has good discriminatory validity. (See Appendix A for specific data.) Additionally, the discriminant validity of each factor is statistically significant. Independent samples *t*-tests (Table 5) and Mann–Whitney U tests (see Appendix A for specific data) were conducted to compare the perceptions of bragging, hypocrisy, and attitudes toward green consumption behavior across different subject groups. The Cohen’s d values obtained from the independent sample *t*-tests represent the effect size, indicating the magnitude of the differences between groups: a larger Cohen’s d value reflects a greater difference. In this study, all Cohen’s d values exceeded 0.8, suggesting a large effect size and indicating that this study possesses sufficient statistical power. Participants exposed to the “Versailles Literature” language style (M_Versailles Literature = 8.867, SD = 2.825, *p* < 0.001) reported significantly lower attitudes toward the poster’s green consumption behavior compared with those exposed to the non-Versailles Literature language style (M_non-Versailles Literature = 12.43, SD = 2.239, *p* < 0.001). Additionally, the results of independent samples *t*-tests for bragging perception and hypocrisy perception indicate that the bragging perception score for the non-Versailles Literature language style (M_non-Versailles Literature = 6.2, SD = 2.964, *p* < 0.001) was significantly lower than that for the “Versailles Literature” language style post (M_Versailles Literature = 11.8, SD = 2.469, *p* < 0.001). Similarly, the hypocrisy perception score for the “non-Versailles Literature” language style post (M_non-Versailles Literature = 5.167, SD = 2.854, *p* < 0.001) was significantly lower than that for the “Versailles Literature” language style post (M_Versailles Literature = 11.5, SD = 2.432, *p* < 0.001). Additionally, controlling for gender, face consciousness, and green consumption awareness, the regression analysis results (Table 6) indicate that exposure to “Versailles Literature” language style posts significantly evokes consumers’ perceptions of bragging (β = 0.71, *p* = 0.000) and hypocrisy (β = 0.783, *p* = 0.000). Both bragging perception (β = −0.582, *p* = 0.000) and hypocrisy perception (β = −0.573, *p* = 0.000) negatively impact consumers’ attitudes toward the poster’s green consumption behavior. Thus, Hypotheses H1, H2a, and H2b are validated.

Finally, following the mediation effect analysis procedure proposed by [84] ([84]), we used [32]’ ([32]) PROCESS method (Model 4, Bootstrap samples = 5000) with bias-corrected non-parametric percentile bootstrapping to test the dual mediating effects of bragging perception and hypocrisy perception. Under a 95% confidence interval, the mediating effect of bragging perception was −0.7758 ([−1.6517, −0.2659], with the confidence interval excluding 0), and the mediating effect of hypocrisy perception was −1.0501 ([−1.9155, −0.2223], with the confidence interval excluding 0). The analysis results show that the indirect effects of both bragging perception and hypocrisy perception are significant. Individuals exposed to the “Versailles Literature” language style posts are influenced by these perceptions, and both bragging and hypocrisy perceptions play parallel mediating roles in the relationship between language style and consumers’ attitudes toward the poster’s green consumption behaviors. These results support Hypotheses H2a and H2b.

### 4.3. Experiment 2

Experiment 2 aimed to explore the impact of language style and the strength of social ties on consumers’ perceptions (bragging perception, hypocrisy perception) and their attitudes toward the poster’s green consumption behaviors. Experiment 2 employed a 2 (Language Style: Versailles Literature vs. Non-Versailles Literature) × 2 (strength of social ties: Stranger vs. Acquaintance) between-subjects design. The participants were randomly recruited from four classes in the same department and grade level at a university in Jiangxi. Thirty students from each class were selected, totaling 120 participants, who were randomly assigned to one of the four groups (28.3% male). The experimental materials consisted of two language style scenarios (Versailles Literature vs. Non-Versailles Literature) and two types of strength of social ties (Stranger vs. Acquaintance). The scenarios were designed to mimic a social media post, and the content was adjusted according to the experimental group. In order to manipulate the strength of social ties, this study added the following guidance at the beginning of the experimental material: “please imagine the following situation: your good friend Li Ming, you have known each other for more than two years, you have a good relationship, often eat and study together (a fellow student you don not know, met once and knew his name is Li Ming, but there is no communication), and he (she) posted the following in the circle of friends. Please read this circle of friends news carefully.” In the non-Versailles Literature condition, the post content was as follows: “Recently, in order to implement green environmental protection principles, I began optimizing my daily consumption choices. I replaced my old electric scooter with a solar-powered, eco-friendly model. Although it cost a little more than a regular one, its clean energy technology makes me feel it is worth the investment. Every time I drive it, I feel that I’m contributing to reducing carbon emissions and hope to encourage more people to pay attention to eco-friendly transportation.” In the Versailles Literature condition, the post content was as follows: “Recently, in order to implement green environmental protection principles, I decided to completely upgrade my daily consumption habits. I replaced my perfectly fine regular electric scooter with an imported, limited-edition solar-powered sports car. Although it cost several times more than a regular car, I’m willing to make that investment for the future of the Earth. Every time I drive it, the attention I receive makes me feel like, ‘Yes, this is the only kind of vehicle that matches my status.’” After reading the respective scenarios, participants were required to complete a questionnaire that measured their perceptions of bragging and hypocrisy, “Versailles Literature” language style, and attitudes toward the poster’s green consumption behaviors. All measurements used a 5-point Likert scale (1 = Strongly Disagree, 5 = Strongly Agree). The experimental materials were controlled for content length and the consistency of the core message, manipulating only the language style factor. A manipulation check was conducted to ensure the successful implementation of the experimental conditions. The experimental procedure and measurement scales were consistent with those in Experiment 1.

#### Experimental Results

Manipulation check results indicated that participants perceived significant differences in language styles; specifically, the measurement values for the “Versailles Literature” language style posts (M_Versailles Literature = 4.17, SD_Versailles Literature = 0.763) were significantly higher than those for the non-Versailles Literature language style posts (M_non-Versailles Literature = 2.37, SD_non-Versailles Literature = 0.920, *p* < 0.001), confirming the successful manipulation of the language style variable.

The demographic information of the study participants is provided in Table 7. SPSS 23.0 was used to evaluate the reliability of the scale, while AMOS 24.0 was applied for confirmatory factor analysis. Table 8 reports the standardized factor loading coefficients, average variance extracted (AVE), composite reliability (CR), and Cronbach’s α values for the variables examined in this study. The results confirm strong convergent validity for bragging perception, hypocrisy perception, and consumers’ attitudes toward green consumption behavior, as all values exceed the 0.5 threshold. The square root of the AVE of each variable is greater than the correlation coefficient between this variable and other variables, and the HTMT value between all variables is lower than the threshold of 0.90, indicating that the scale in this study has good discriminatory validity. (See Appendix A for specific data.) To compare perceptions of bragging, hypocrisy, and attitudes toward green consumption behavior across different participant groups, independent samples *t*-tests (Table 9) and Mann–Whitney U tests (See Appendix A for specific data) were performed. Independent samples *t*-test results for consumers’ attitudes toward the poster’s green consumption behaviors show that there was no significant difference in attitudes between participants who saw posts from strangers and those who saw posts from acquaintances in the “non-Versailles Literature” language style condition (M_acquaintance = 12.3, M_stranger = 12.53, t(58) = −0.57, *p* > 0.05). However, for participants who saw posts in the “Versailles Literature” language style, attitudes toward the poster’s green consumption behaviors were significantly different between those who saw posts from strangers and those who saw posts from acquaintances (M_acquaintance = 9.267, M_stranger = 6.20, t(58) = 4.464, *p* < 0.05). Independent samples *t*-test results for the moderating effects of language style and relationship strength on bragging perception and hypocrisy perception show that participants who saw posts from strangers in the “non-Versailles Literature” condition had significantly higher bragging perception than those who saw posts from acquaintances (M_acquaintance = 5.20, M_stranger = 6.37, t(58) = −3.212, *p* < 0.05). However, there was no significant difference in hypocrisy perception (M_acquaintance = 5.13, M_stranger = 5.733, t(58) = −1.723, *p* > 0.05). For participants who saw posts in the “Versailles Literature” language style, those who saw posts from strangers had significantly higher bragging perception (M_acquaintance = 8.60, M_stranger = 12.03, t(58) = −4.906, *p* < 0.05) and hypocrisy perception (M_acquaintance = 8.80, M_stranger = 12.27, t(58) = −4.744, *p* < 0.05) compared with those who saw posts from acquaintances.

Furthermore, the discriminant validity of each factor was statistically significant. The statistical analysis results show that the main effect of language style on consumers’ attitudes toward the poster’s green consumption behavior was significant (F(1,120) = 137.175, *p* < 0.001), further validating Hypothesis 1. The main effect of the strength of social ties (stranger vs. acquaintance) on consumers’ attitudes toward the poster’s green consumption behavior was also significant (F(1,120) = 12.552, *p* = 0.001). The analysis of variance results indicate that the interaction between language style and relationship strength significantly influenced consumers’ attitudes toward the poster’s green consumption behavior (F(1,120) = 17.027, *p* < 0.001). A simple effects analysis was further conducted, as shown in Table 10. When the language style of the post content was non-Versailles Literature, the strength of social ties had no statistically significant effect on consumers’ attitudes toward the poster’s green consumption behavior (*p* > 0.05). However, when the language style of the post content was “Versailles Literature”, the strength of social ties had a statistically significant effect on consumers’ attitudes toward the poster’s green consumption behavior in all pairwise comparisons (*p* < 0.05).

The moderating effect of the strength of social ties on the relationship between language style and consumers’ attitudes toward the poster’s green consumption behavior was analyzed using Bootstrapping (PROCESS Model 7). The results show that the strength of social ties effectively moderates the relationship between the poster’s language style and consumers’ attitudes through bragging perception and hypocrisy perception; specifically, language style significantly impacts consumers’ perceptions of bragging (β = 4.5019, 95% CI [3.6199, 5.3839]) and hypocrisy (β = 5.0382, 95% CI [4.1572, 5.9192]). The interaction between language style and relationship strength significantly influences both bragging perception (β = 1.9574, 95% CI [0.1770, 3.7378], excluding 0) and hypocrisy perception (β = 2.6161, 95% CI [0.8377, 4.3945], excluding 0). Additionally, after controlling for the mediating variable of bragging perception, the direct effect of language style on consumers’ attitudes toward the poster’s green consumption behavior remained significant (Direct effect = −1.9050, 95% CI [−2.7938, −1.0161]). The strength of social ties effectively moderated the relationship between language style and consumers’ attitudes through bragging perception (Indirect effect = −1.1542, 95% CI [−2.3094, −0.1123]). Similarly, after controlling for the mediating variable of hypocrisy perception, the direct effect of language style on consumers’ attitudes toward the poster’s green consumption behavior was significant (Direct effect = −1.3730, 95% CI [−2.2636, −0.4823]). The strength of social ties effectively moderated the relationship between language style and consumers’ attitudes through hypocrisy perception (Indirect effect = −1.6617, 95% CI [−2.9262, −0.4927]). Thus, bragging perception and hypocrisy perception play partial mediating roles in the influence of language style and strength of social ties on consumers’ attitudes toward the poster’s green consumption behavior.

### 4.4. Experiment 3

This experiment employed a 2 (language style: Versailles Literature vs. non-Versailles Literature) × 2 (strength of social ties: strangers vs. acquaintances) between-subjects factorial design. A total of 124 participants were recruited from a university in China. To closely replicate real-world conditions, participants were asked to browse a simulated social media platform for five minutes before the experiment began. The platform featured multiple “background posts” covering topics such as celebrity gossip, food exploration, and campus news, with some posts accompanied by short videos and interactive comments, simulating a realistic social media feed. Considering the possible contextual inconsistency caused by direct guidance, unlike previous experiments that manipulated the strength of social ties through textual prompts, this study utilized the inherent characteristics of the platforms themselves (WeChat vs. Weibo) as the manipulation medium. To avoid contextual inconsistencies introduced by explicit prompts, two distinct user interfaces (WeChat and Weibo) were designed. The interfaces maintained consistency in avatar, username, number of likes, and number of comments. The two posts were matched in terms of informational content, font, and word count. This approach allowed for the manipulation of the strength of social ties by leveraging platform attributes.

Participants were then randomly assigned to one of the four experimental conditions and asked to continue browsing the content on the simulated social media platform. Participants in the acquaintance group were asked to read a green consumption update posted on Moments by a WeChat friend who “frequently posts lifestyle content”, while those in the stranger group were asked to read a green consumption update posted on Weibo by an unfollowed Weibo user who “frequently posts lifestyle content”. To enhance ecological validity and align with the linguistic norms of real social media platforms, the post texts incorporated common stylistic features and popular hashtags. Non-Versailles Literature version: “After work today, I stopped by that imported organic supermarket and picked up some additive-free household items. The quality seems pretty good. The cashier even said I made some great choices, which made me quite happy. I have recently started trying a low-carbon lifestyle and found it surprisingly convenient. My friends also said I’m doing a good job. I believe environmental protection is something everyone can contribute to—starting with small actions and doing what we can really matters. #GreenLiving #EcoCheckIn #LifeIsNotEasy”. Versailles Literature version: “After work today, I once again went to that imported organic supermarket—after all, the everyday items in regular stores are just too ‘ordinary’ for me. I ended up buying a bunch of additive-free household goods. The cashier even praised me for having such great taste—it was almost a bit embarrassing. I have recently started trying a low-carbon lifestyle, and surprisingly, it is even more convenient than before. My friends all say I’m ‘so particular,’ but honestly, I just did a few small things—nothing special. #GreenLiving #EcoCheckIn #LifeIsNotEasy”.

Subsequently, participants were asked to evaluate their perceptions of the language style used in the post with questions such as “Do you think the description of the content of this post is natural and simple or exaggerated and showy?” “Do you think this text represents a “Versailles Literature” language style? (“Versailles Literature” language style refers to the use of apparent modesty, complaint, or self-deprecation to subtly convey one’s superior status or achievements).” They were also asked to assess their perceptions of the strength of social relationships associated with the platform: “I believe WeChat/Weibo is a strong-tie social platform” and “Most people I interact with on WeChat/Weibo are those I know well.” Participants then rated their attitudes toward the poster’s green consumption behavior, as well as their perceptions of the bragging and hypocrisy, using the same measurement scales as in Experiment 1. To control for potential confounding factors related to perceived differences in identity hierarchy, follower count, or prior green consumption behavior frequency, participants were also asked to report the following: “Do you think the person who posted this content is similar to you in terms of characteristics such as age or occupation?” “Do you think this user has many followers?” “In the past month, how many times have you engaged in the following green consumption behaviors? Please answer based on your actual experience.”

#### Experimental Results

The manipulation check results indicated significant differences in participants’ perceptions of language style. In the two “non-Versailles Literature “conditions, 93.55% of participants perceived the post content as natural and simple. In contrast, 90.32% of participants in the two “Versailles literature” conditions perceived the content as exaggerated and showy. Furthermore, the measured score for “Versailles Literature” posts was significantly higher than that for “non-Versailles Literature“ posts (M_Versailles = 3.89, SD = 0.832; M_Non-Versailles = 1.71, SD = 0.755; *p* < 0.001). Similarly, participants’ perceptions of relationship closeness significantly differed based on platform type. Specifically, posts on WeChat were associated with significantly higher perceived relational closeness than those on Weibo (M_WeChat = 3.90, SD = 0.844; M_Weibo = 1.87, SD = 0.820; *p* < 0.001).

The demographic characteristics of the participants are presented in Table 11. We conducted independent samples *t*-tests (Table 12) and Mann–Whitney *U* tests (See Appendix A for specific data). The analyses revealed no significant differences in attitudes toward the poster’s green consumption behavior, bragging perception, or hypocrisy perception between participants who viewed a “non-Versailles Literature” post from a stranger and those who viewed a “non-Versailles Literature “post from an acquaintance.

In contrast, significant differences were observed among participants who viewed “Versailles Literature” language style posts: those who viewed such a post from a stranger reported significantly different attitudes toward the poster’s green consumption behavior, as well as significantly higher levels of bragging perception and hypocrisy perception compared with those who viewed the same post from an acquaintance. A regression analysis was conducted on the “Versailles Literature” condition, with identity hierarchy, prior green consumption behavior frequency, follower count, face consciousness, and green consumption awareness included as control variables (Table 13). The results indicated that exposure to “Versailles Literature” post content significantly increased consumers’ perceptions of bragging (β = 0.703, *p* = 0.000) and hypocrisy (β = 0.653, *p* = 0.000). Furthermore, both bragging perception (β = −0.740, *p* = 0.000) and hypocrisy perception (β = −0.813, *p* = 0.000) negatively influenced participants’ attitudes toward the poster’s green consumption behavior. ANOVA results revealed a significant main effect of language style on consumers’ attitudes toward the poster’s green consumption behavior (F(1,120) = 103.148, *p* = 0.000), providing further support for Hypothesis 1. There was also a significant main effect of strength of social ties (F(1,120) = 15.306, *p* = 0.000). Moreover, the interaction between language style and strength of social ties significantly influenced participants’ attitudes toward the poster’s green consumption behavior (F(1,120) = 19.242, *p* = 0.000).

A simple effects analysis was conducted to further interpret the interaction effect (Table 14). When the language style of the post was non-Versailles, there were no statistically significant differences in consumers’ attitudes toward the poster’s green consumption behavior across levels of strength of social ties (*p* > 0.05). However, when the language style was Versailles, all pairwise comparisons across strength of social ties levels showed statistically significant differences (*p* < 0.05). Finally, a moderated mediation model was tested. Language style was entered as the independent variable (non-Versailles = 0, Versailles = 1), with bragging perception and hypocrisy perception as mediating variables. Strength of social ties was included as a moderator (acquaintance = 0, stranger = 1). Green consumption awareness, face consciousness, identity hierarchy, perceived follower count, and prior green consumption behavior frequency were included as control variables. Using Models 4 and 1 of the PROCESS macro in SPSS, a bootstrapping procedure with 5000 resamples and a 95% confidence interval was employed to examine the mediation and moderation effects. The results indicated that both bragging perception (β = −2.973, 95% CI = [−4.132, −2.076]) and hypocrisy perception (β = −3.186, 95% CI = [−4.346, −2.346]) had significant mediating effects, as their confidence intervals did not include zero. Strength of social ties served as a significant moderator in the model (Table 15). Specifically, for the mediating variable, bragging perception, the interaction effect between language style and strength of social ties was significant (β = 2.144, 95% CI = [0.359, 3.930], CI does not include 0). Similarly, the interaction effect was also significant for hypocrisy perception (β = 3.374, 95% CI = [1.593, 5.155], CI does not include 0). These results suggest that the interaction between language style and strength of social ties significantly influences both mediators, supporting the presence of a moderated mediation effect. These findings indicate that H3a and H3b were supported; the specific results are shown in Table 15 below.

## 5. Discussions

### 5.1. Theoretical Contributions

This study extensively explored the impact of language style (Versailles Literature and non-Versailles Literature) on consumers’ attitudes towards the green consumption behavior of posters based on the theory of self-presentation. It enriches the application and understanding of the “Versailles Literature” language style in the field of green consumption behavior. In the context of green consumption, “Versailles Literature” language style is a rhetorical means of irony, exaggeration, contrast, and so on, which disguises its own achievements in green consumption behavior through superficial humility, complaint, or self-mockery. Previous studies have focused on social psychology, brand marketing, tourism marketing, and corporate investment ([69]; [29]; [13]; [24]; [81]). This study explores the impact mechanism of “Versailles Literature” language style on green consumption behavior attitude on social media and highlights the unique role of language style in shaping consumer attitudes. Compared with “non-Versailles Literature”, “Versailles Literature” language style is regarded as a hypocritical self-display, thus reducing the audience’s positive attitude towards the green consumption behavior of the poster. This expands the research on self-display strategies on social media and the “Versailles Literature” language style in the field of green consumption ([42]), and provides inspiration for communication strategies utilized by enterprises and marketers when promoting green consumption.

Secondly, this study reveals the internal mechanism through which the “Versailles Literature” language style affects individuals’ attitudes towards green consumption behavior. Existing studies have explored the single, chain, and parallel mediators of the “Versailles Literature” language style, such as perceived sincerity, reviewer favorability, benign jealousy, amusement, annoyance, ostentation, and sincerity ([69]; [13]; [60]; [54]). Unlike previous studies on cognitive or emotional mediators, this study explores the mediating roles of bragging and hypocrisy perceptions on green consumption, a specific situation with moral sensitivity. Compared with “non-Versailles Literature” language style, “Versailles Literature” language style increases the audience’s perception of bragging and hypocrisy by the poster, thus reducing the audience’s positive attitude towards the poster’s green consumption behavior. This study echoes the central idea of the importance of motivation perception in self-presentation theory. Elucidating the indirect influence of these perceptions on attitudes toward green consumption behavior offers a new perspective on how consumers form negative evaluations of consumption behaviors in social interactions.

Unlike traditional studies on green consumption behavior that focus primarily on consumers’ intrinsic motivations and socio-cultural backgrounds, and traditional green bleaching behavior research that focuses mainly on the impact of false environmental protection publicity strategies at the enterprise level on consumer cognition and purchase behavior ([53]; [70]; [18]), this study explores the psychological mechanism of the “green flaunting” phenomenon at the individual level from the perspective of individual consumer language expression, providing an important supplement to studies on green bleaching behavior from the micro-individual perspective. Combined with the perception theory in social psychology, this study discusses the impact of language style and social relationships on consumer attitudes and reveals the multi-dimensional impact of the complex interweaving of language and social interaction on the formation of green consumption attitudes. Through these theoretical insights, this study provides additional avenues for future research on green consumption behavior and fosters interdisciplinary integration between consumer behavior and social psychology.

### 5.2. Managerial Implications

Based on the background of social media and digital platforms, this study focuses on the impact of online language style and the strength of social ties on individuals’ attitudes toward green consumption behavior. The findings reveal that when consumers encounter green consumption information presented in a “Versailles Literature” language style, their attitudes change significantly, and the interaction between language style and social relationships has a notable impact on their recognition and support of others’ green consumption behaviors. In the context of strong ties (such as private social platforms such as WeChat), the “Versailles Literature” language style may be accepted and elicit a positive response from consumers, as they have a higher level of trust and emotional connection with the poster. However, in the context of weak ties (such as public social platforms such as Weibo), excessive exaggeration and boastful language may trigger perceptions of boasting and hypocrisy, which in turn weaken consumers’ attitudes toward brands’ green consumption behavior.

Therefore, when promoting green consumption behavior, businesses need to tailor their language style according to the strength of social relationships on different platforms. On platforms with strong ties (such as WeChat), it is acceptable to use a slightly exaggerated or boastful “Versailles Literature” language style, as this helps strengthen the emotional bond with consumers and enhances the appeal of the brand’s message. However, on social platforms with weak ties (such as Weibo), businesses should avoid using “Versailles Literature” language style and instead adopt a more sincere and transparent communication style, conveying their green actions and sustainability commitments genuinely, rather than relying on exaggerated marketing or “greenwashing” tactics to attract consumer attention. Genuine communication and tangible actions are the foundations for building brand trust, and can effectively reduce consumers’ perceptions of hypocrisy and enhance their attitudes toward green consumption.

Moreover, businesses should pay more attention to the moderating effect of the strength of social ties in social media marketing, particularly in building close, trusting relationships with consumers to minimize the negative impact of language style. To achieve this, businesses should ensure transparency and authenticity in their language while also reinforcing brand credibility through verifiable green practices, such as product certifications and environmental reports. By doing so, companies can effectively increase consumer recognition of green consumption behaviors, enhance their willingness to engage in green consumption, and promote the long-term development and sustainability of the brand.

This study was conducted against the background of Chinese culture, and the research object “Versailles Literature” language style, as a unique social network cultural phenomenon in China, has obvious cultural particularity. The linguistic characteristics, expression, consumer understanding, and reaction to the “Versailles Literature” language style are deeply influenced by Chinese cultural traditions, social values, and the online cultural environment. The phenomenon of “humblebragging” on Western social media bears similarities to the “Versailles Literature” language style. Consumers in the region may have the same understanding and emotional response to similar language styles; thus, the findings of this study may provide a useful reference for understanding similar language phenomena in a cross-cultural context.

### 5.3. Research Limitations and Future Research Directions

Although this study provides in-depth insights into the influence of language style and strength of social ties on consumers’ attitudes toward the green consumption behaviors of posters, several limitations remain. Firstly, although participants who saw non-Versailles Literature language posts posted by strangers perceived a higher level of bragging than those who saw non-Versailles Literature language posts posted by acquaintances, no significant difference was observed in consumers’ attitudes toward the green consumption behaviors of the posters. This result may be associated with the particularity of the sample and the setting of the situation. For example, participants may interpret the posts in different interpersonal contexts, and attitudes toward green consumption behavior may not be significantly affected by the perception of bragging. In addition, the moderating variable, the proximity of the strength of social ties, may not affect the dependent variable; this may also be one of the reasons for the insignificant results. Future research could further explore different types of relationships and more diverse settings to gain a deeper understanding of how these relationships work.

Secondly, the participants consisted of a small number of students from a single university in southern China; however, there are significant differences in economic development levels, cultural traditions, and consumption concepts in different regions. Additionally, the participants were between 18 and 25 years old. Consumers in this age group usually have strong social media use ability, high sensitivity to popular online culture, and are in the critical period of consumption concepts and value formation. Consumers of different ages may have significant differences in terms of social media use habits, language understanding ability, and green consumption cognitive levels. Thus, the findings of this study may be more relevant to younger people, with their relevance to other age groups or groups of people from different backgrounds being unclear. Future studies should consider expanding the sample to include individuals of more diverse ages, occupations, and social backgrounds in order to improve the generalizability and external validity of the results.

Thirdly, this study focused on only two language styles—“Versailles Literature” and “non-Versailles Literature”. Future research could explore a more diverse range of language styles and investigate their potential impacts on individuals’ attitudes toward the green consumption behaviors of posters. Finally, the findings of this study are limited by the specific design of the selected platform. This study relied on a single Chinese social media platform whose design and functionality may differ significantly from those of other social media platforms, such as Facebook or Instagram, which are global social media platforms. This may limit the generalizability of this study’s conclusions to other digital communication environments. Furthermore, different social media platforms may exert varying influences on consumers’ language styles and attitudes. For example, visually oriented platforms (such as Xiaohongshu and Douyin) may have different effects on language style compared with text-based social media platforms (such as Weibo and Zhihu). Future research could examine the moderating role of platform type on the effectiveness of language style on social media marketing.

## 6. Research Conclusions

This study explored the impact of language style (Versailles Literature vs. non-Versailles Literature) and the strength of social ties on consumers’ attitudes toward the poster’s green consumption behavior, and examined the mediating roles of bragging perception and hypocrisy perception. The results showed that the “Versailles Literature” language style significantly reduced consumers’ positive attitudes toward the poster’s green consumption behaviors while increasing their perceptions of bragging and hypocrisy. In contrast, the “non-Versailles Literature” language style had a smaller effect on attitudes toward green consumption behaviors, bragging perception, and hypocrisy perception. [8] ([8]) mentioned that low-key consumption signals (such as high-end brands of consumer goods without obvious labels) can only be identified by groups with cultural capital. Similarly, the “Versailles Literature” language style conveys its own social status signals through subtle language styles. This leads to others’ judgment of the poster’s identity and motives. Furthermore, the strength of social ties played a crucial moderating role in this process, particularly when participants saw posts from strangers using the “Versailles Literature” language style, where they exhibited stronger perceptions of bragging and hypocrisy. Further analysis revealed that bragging perception and hypocrisy perception partially mediated the influence of language style and strength of social ties on consumers’ attitudes toward green consumption behaviors; specifically, bragging perception and hypocrisy perception indirectly affected attitudes toward green consumption behaviors by intensifying consumers’ negative perceptions. This study also found that the interaction between language style and strength of social ties significantly moderated consumers’ perceptions of bragging and hypocrisy, and through the mediating roles of these perceptions, further influenced consumers’ attitudes toward green consumption behaviors. [80] ([80]) pointed out that changes in psychological distance can trigger a transformation in an individual’s processing pattern, thereby affecting their abstraction or concrete construction of information. Combined with the findings of this study, the Versailles Literature language style may enhance consumers’ psychological distance towards the poster, thus triggering abstract processing of green consumption behavior; that is, they no longer focus on the environmental protection significance of the behavior itself, but are more likely to interpret it as symbolic behavior of “bragging” or “hypocrisy”. This transformation of the processing mode further intensifies the negative perception and reduces consumers’ attitudes towards green consumption behaviors.

## Figures and Tables

**Figure 1 behavsci-15-00968-f001:**
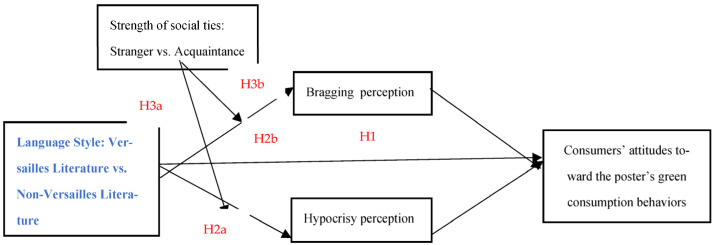
The conceptual model for the influence of language style on consumers’ attitudes toward the poster’s green consumption behaviors.

**Table 1 behavsci-15-00968-t001:** Comparison of the characteristics and influence of different language styles.

Characteristic Dimension	Versailles Literature	Direct Bragging	Virtue Signaling
Apply perspective	third-person perspective	first-person perspective	collective identity expression
Comparison strategy	reverse contrast, desire to rise first and restrain first	positive comparison or independent statement	contrast with unethical behavior
Language characteristics	ostensibly humble, ostensibly ostentatious	explicit self-praise	values manifesto
Social influence	arouse audience disgust and reduce social support	enhance impression and trigger jealousy	enhance group identity and may trigger moral licensing effect
Related literature	[67] ([67]); [69] ([69])	[68] ([68]); [29] ([29])	[39] ([39]); [43] ([43])

**Table 2 behavsci-15-00968-t002:** Overview of this study.

Experiment		Purpose	Method	Sample Status
Experiment 1	Main effect, Mediating effect	To verify the direct impact of “Versailles Literature” language style on green consumption behavior and its mediating mechanism	Used a one-way intergroup design (language Style: Versailles Literature vs. non-Versailles Literature)	N = 60 35% male
Experiment 2	Moderating effect	Verify the moderating effect of the strength of social ties	Used a two-factor design of 2 (language style: Versailles Literature vs non-Versailles Literature) × 2 (strength of social ties: stranger vs. acquaintance)	N = 120 28.3% male
Experiment 3	Moderating effect	Platform differentiation design is used to control the strength of social ties and enhance the moderating effect of the strength of social ties	Used a two-factor design of 2 (language style: Versailles Literature vs. non-Versailles Literature) × 2 (strength of social ties: stranger vs. acquaintance)	N = 124 25.8% male

**Table 3 behavsci-15-00968-t003:** Demographic characteristics of Experiment 1.

Parameter	Category	Frequency	Percentage (%)
Gender	Male	21	35
	Female	39	65
Age	Under 18	13	21.7
	19–25	44	73.3
	26–30	3	5
Education	Undergraduate	57	95
	Postgraduate and above	3	5
Occupation	Student	60	100

**Table 4 behavsci-15-00968-t004:** The reliability and convergent validity of the mediating variables and the dependent variables of Experiment 1.

Variables	Laten Variables	S.E.	C.R.	*p*	Std. Factor Loading	AVE	CR	Cronbach’s α
BP	BP1				0.867	0.860	0.948	0.949
	BP2	0.101	11.806	***	0.968			
	BP3	0.108	11.177	***	0.945			
HP	HP1				0.901	0.897	0.963	0.963
	HP2	0.079	13.265	***	0.956			
	HP3	0.079	14.527	***	0.983			
LS	LS1				0.889	0.851	0.944	0.945
	LS2	0.089	12.349	***	0.969			
	LS3	0.094	10.744	***	0.907			

*** *p* < 0.001.

**Table 5 behavsci-15-00968-t005:** Independent samples *t*-test results for mediating variables and dependent variables of Experiment 1.

Variables	Variables	M	SD	*p*	S2pooled	Cohen’s d
attitudes toward the poster’s green consumption behavior	non-Versailles Literature	12.43	2.239	0.000	6.497	1.399
	Versailles Literature	8.867	2.825			
bragging perception	non-Versailles Literature	6.2	2.964	0.000	7.441	2.053
	Versailles Literature	11.8	2.469			
hypocrisy perception	non-Versailles Literature	5.167	2.854	0.000	7.029	2.389
	Versailles Literature	11.5	2.432			

**Table 6 behavsci-15-00968-t006:** Regression analysis of mediating variables and dependent variables of Experiment 1.

Dependent Variable	Variables	β	t	*p*	VIF	R2
bragging perception	Constant		1.995	0.051		0.532
	Language style	0.710	6.739	0.000	1.303	
hypocrisy perception	Constant		2.521	0.015		0.606
	Language style	0.783	8.100	0.000	1.303	
attitudes toward the poster’s green consumption behavior	Constant		3.423	0.001		0.429
	Bragging Perception	−0.582	−5.282	0.000	1.170	
	Constant		3.560	0.001		0.424
	Hypocrisy Perception	−0.573	−5.207	0.000	1.157	

**Table 7 behavsci-15-00968-t007:** Demographic characteristics of Experiment 2.

Parameter	Category	Frequency	Percent (%)
Gender	Male	34	28.3
	Female	86	71.7
Age	Under 18	15	12.5
	19–25	103	85.8
	26–30	2	1.7
Education	Undergraduate	120	100
Occupation	Student	120	100

**Table 8 behavsci-15-00968-t008:** The reliability and convergent validity of the mediating variables and the dependent variables of Experiment 2.

Variables	Laten Variables	S.E.	C.R.	*p*	Std. Factor Loading	AVE	CR	Cronbach’s α
BP	BP1				0.809	0.800	0.923	0.921
	BP2	0.1	12.727	***	0.929			
	BP3	0.107	12.954	***	0.939			
HP	HP1				0.959	0.866	0.951	0.950
	HP2	0.052	18.287	***	0.895			
	HP3	0.047	21.995	***	0.937			
LS	LS1				0.917	0.876	0.955	0.954
	LS2	0.062	18.232	***	0.934			
	LS3	0.056	19.627	***	0.956			

*** *p* < 0.001.

**Table 9 behavsci-15-00968-t009:** Independent samples *t*-test results for mediating variables and dependent variables of Experiment 2.

Variables	Variables	M	SD	t	*p*
attitudes toward the poster’s green consumption behavior	non-Versailles Literature Acquaintance group	12.30	1.84	−0.570	0.571
	non-Versailles Literature stranger group	12.53	1.28		
	Versailles Literature acquaintance group	9.267	3.07	4.464	0.000
	Versailles Literature stranger group	6.20	2.17		
bragging perception	non-Versailles Literature acquaintance group	5.20	1.95	−2.251	0.028
	non-Versailles Literature stranger group	6.37	2.06		
	Versailles Literature acquaintance group	8.60	3.35	−4.906	0.000
	Versailles Literature stranger group	12.03	1.87		
hypocrisy perception	non-Versailles Literature acquaintance group	5.13	1.83	−1.200	0.235
	non-Versailles Literature stranger group	5.733	2.03		
	Versailles Literature acquaintance group	8.80	3.43	−4.744	0.000
	Versailles Literature stranger group	12.27	2.07		

**Table 10 behavsci-15-00968-t010:** Pairwise comparisons of language style and the strength of social ties of Experiment 2.

Dependent Variable: Attitudes Toward the Poster’s Green Consumption Behavior							
Language Style	(I) Strength of Social Ties	(J) Strength of Social Ties	Mean Difference (I-J)	Std. Error	Sig. b	95% Confidence Interval for Difference b	
						Lower Bound	Upper Bound
non-Versailles Literature	acquaintance	stranger	−0.233	0.565	0.681	−1.353	0.887
	stranger	acquaintance	0.233	0.565	0.681	−0.887	1.353
Versailles Literature	acquaintance	stranger	3.067 *	0.565	0.000	1.947	4.187
	stranger	acquaintance	−3.067 *	0.565	0.000	−4.187	−1.947

Based on estimated marginal means. * The mean difference is significant at the 0.05 level. b Adjustment for multiple comparisons: Least Significant Difference (equivalent to no adjustments).

**Table 11 behavsci-15-00968-t011:** Demographic characteristics of Experiment 3.

Parameter	Category	Frequency	Percent (%)
Gender	Male	32	25.8
	Female	92	74.2
Age	Under 18	1	0.8
	19–25	121	97.6
	26–30	2	1.6
Education	Undergraduate	110	88.7
	Postgraduate and above	14	11.3
Monthly Income	under CNY 1000	63	50.8
	CNY 1000–2000	51	41.1
	CNY 2000–3000	8	6.5
	CNY 6000–10,000	2	1.6

**Table 12 behavsci-15-00968-t012:** Independent samples *t*-test results for mediating variables and dependent variables of Experiment 3.

Variables	Variables	M	SD	t	*p*
attitudes toward the poster’s green consumption behavior	non-Versailles Literature acquaintance group	12.16	1.92	−0.360	0.720
	non-Versailles Literature stranger group	12.35	2.30		
	Versailles Literature acquaintance group	9.81	2.95	5.518	0.000
	Versailles Literature stranger group	6.42	1.73		
bragging perception	non-Versailles Literature acquaintance group	4.52	1.75	−0.870	0.388
	non-Versailles Literature stranger group	5.03	2.56		
	Versailles Literature acquaintance group	9.19	2.61	−3.422	0.001
	Versailles Literature stranger group	11.55	2.80		
hypocrisy perception	non-Versailles Literature acquaintance group	4.61	1.71	−0.728	0.470
	non-Versailles Literature stranger group	5.03	2.71		
	Versailles Literature acquaintance group	8.42	2.98	−5.232	0.000
	Versailles Literature stranger group	12.00	2.38		

**Table 13 behavsci-15-00968-t013:** Regression analysis of mediating variables and dependent variables of Experiment 3.

Dependent Variable	Variable	β	t	*p*	VIF	R^2^
bragging perception	Constant		−0.563	0.575		0.568
	language style	0.703	11.430	0.000	0.703	
hypocrisy perception	Constant		0.440	0.660		0.536
	language style	0.653	10.235	0.000	0.653	
attitudes toward the poster’s green consumption behavior	Constant		6.775	0.000		0.627
	bragging perception	−0.740	−12.533	0.000	−0.740	
	Constant		8.620	0.00		0.707
	hypocrisy perception	−0.813	−15.250	0.000	−0.813	

**Table 14 behavsci-15-00968-t014:** Pairwise comparisons of language style and the strength of social ties of Experiment 3.

Dependent Variable: Attitudes Toward the Poster’s Green Consumption Behavior							
Language Style	(I) Strength of Social Ties	(J) Strength of Social Ties	Mean Difference (I-J)	Std. Error	Sig. b	95% Confidence Interval for Difference	
						Lower Bound	Upper Bound
non- Versailles Literature	acquaintance	stranger	−0.194	0.577	0.738	−1.336	0.949
	stranger	acquaintance	0.194	0.577	0.738	−0.949	1.336
Versailles Literature	acquaintance	stranger	3.387 *	0.577	0.000	2.244	4.530
	stranger	acquaintance	−3.387 *	0.577	0.000	−4.530	−2.244

Based on estimated marginal means. * The mean difference is significant at the 0.05 level. b Adjustment for multiple comparisons: Least Significant Difference (equivalent to no adjustments).

**Table 15 behavsci-15-00968-t015:** Test of the moderating effect on the strength of social ties of Experiment 3.

Mediation Variables	Moderating Variables	Effect	SE	*p*	LLCI	ULCI
bragging perception	acquaintance	4.345	0.647	0.000	3.063	5.628
	stranger	6.490	10.228	0.000	5.233	7.746
hypocrisy perception	acquaintance	3.448	0.646	0.000	2.169	4.727
	stranger	6.822	0.633	0.000	5.568	8.075

## Data Availability

The datasets generated during and/or analyzed during the current study are available from the corresponding author on reasonable request.

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
