# Peer review of "How Will I Evaluate Others? The Influence of “Versailles Literature” Language Style on Social Media on Consumer Attitudes Towards Evaluating Green Consumption Behavior"

_behavsci, 2025, doi:10.3390/bs15070968_

Round 1
Reviewer 1 Report (Previous Reviewer 2)
Comments and Suggestions for Authors
Major issues:
- Justifications are expected for the claim on Lines 238-239: “Versailles Literature” embodies two primary motivations: seeking 238 affection and gaining respect.
- It seems that H2a and H2b are based on H1, then why did the authors use two separate hypotheses? If H1 and H2 are different, please rephrase them.
- An overview of the experiments in Section 4 will prepare readers for the coming experiments.
- Regarding the experiments, the authors should provide more details regarding data collection (e.g. participant recruitment; experiment tool/platform), data processing and data analysis.
- In the design of Exp 2, how was the factor of Social ties manipulated?
- Too many tables are presented in the article. The authors may consider using some figures to increase the readability. Some tables can be included in supplementary materials.
- Conclusion should come after discussion.
- The authors should connect their findings to the theories and studies discussed earlier in the paper and provide a clearer interpretation of the results. The current discussion section is underdeveloped and lacks depth.
Minor issues:
- The authors are recommended to pay attention to the formatting. For example, the references should appear before the full stop on Lines 58 and 63.
- Please proofread the article carefully. Lines 241, 246, 267, 272 (and more): a space is needed before “language”.
- The formatting of the reference list should be consistent. For example, References 28 and 39 provide full names of authors while others only include initials for given names.
- The authors may revise some long sentences to increase the readability of the article. For example, Lines 251-254 is difficult to comprehend.
- Please do not Chinese punctuation marks. For example, brackets from Line 417.
- Since the same materials were used in the formal experiment, there’s no point restating the two posts on Lines 400-412.
- Section 4.1.2: Since there are several statements, the authors may consider listing them in a table.
- Table 1: No need to include the Cumulative Percentage (%)
Author Response
Thank you for your constructive feedback. We have carefully considered each of your suggestions and made revisions accordingly. Below are detailed responses to each of your comments.
Comment 1: Justifications are expected for the claim on Lines 238-239: “Versailles Literature” embodies two primary motivations: seeking 238 affection and gaining respect.
Response 1:
Thank you for your suggestions regarding your comment about the motivation behind the “Versailles Literature” in lines 238-239, which lacks sufficient justification. We added a relevant theoretical basis and empirical research to the manuscript, supplemented Leary and Kowalski (1990)’s impression management framework, and provided a detailed explanation of the correlation between “Versailles Literature” and self-presentation theory, providing solid empirical support for the theoretical proposition. Thank you for your valuable feedback. Your suggestions have been very helpful in improving the quality of our manuscript, and we look forward to your further review. The detailed revised draft is on page 6 line 265-281, and the excerpt is as follows:
According to Goffman’s (1949) self-presentation theory, individuals often adopt specific linguistic and behavioral strategies during social interactions to shape how they are perceived by others and gain the respect of others. The classical two-factor model proposed by Leary and Kowalski (1990) shows that impression management is com-posed of two core components: impression motivation and impression construction. Studies by Sezer et al. (2018) and Baumeister (1982) found that “Versailles Literature” has a dual role. Individuals highlight their strengths and achievements through con-spicuous behavior, which aims to build cognitive competence and win the respect and recognition of others. Individuals can also use strategies such as humble expression and moderate complaint to create a prosocial image, which can not only stimulate others’ sympathetic response, but also convey intimacy signals and trust hints, thus promoting the establishment of loving relationships. This reflects that two core views of individual self-presentation are highly similar: One is to seek liking motivation; that is, to obtain the emotional identity of others by shaping an attractive and affinity self-image (Hol-lenbaugh, 2021). The second is to respect acquisition motivation, which is to build au-thoritative awareness and win the respect of others by demonstrating personal abilities, achievements, and values (Bergsieker, 2010; Ma, 2023). Therefore, “Versailles Literature” is a special expression phenomenon in the era of social media, and is often used to seek self-improvement and win the recognition and love of others.
- Bergsieker, H. B., Shelton, J. , & Richeson, J. A. (2010). To be liked versus respected: Divergent goals in interracial interactions. Journal of personality and social psychology, 99(2), 248.
- Hollenbaugh, E. E. H. E. E. (2021). Self-presentation in social media: Review and research opportunities. Review of communication research, 9.
- Leary, M. , & Kowalski, R. M. (1990). Impression management: A literature review and two-component model. Psychological bulletin, 107(1), 34.
Comment 2: It seems that H2a and H2b are based on H1, then why did the authors use two separate hypotheses? If H1 and H2 are different, please rephrase them.
Response 2:
Thank you very much for your valuable feedback on the research hypothesis section. After rethinking, we adopted a hierarchical hypothesis structure to express the research logic more clearly: H1 as the overall effect hypothesis, predicting the direct impact of “Versailles Literature” language style on consumers’ negative attitudes; H2 is used as a mechanism hypothesis to explore the psychological mechanism of this effect, including two parallel mediating pathways (hypocrisy perception and bragging perception). This maintains the independence of the assumptions and clarifies the logical relationship between them, thus providing a clearer framework for subsequent statistical analysis. We believe that these adjustments will make the assumptions clearer and more reasonable, enhancing the logical coherence of the article. Thank you again for your careful review, and we look forward to your further feedback. The detailed revised draft is on page 7 line 330-338, and the excerpt is as follows:
H1: The “Versailles Literature” language style is more likely to trigger negative consumer attitudes towards the green consumption behavior of the sender than a non-Versailles language style.
H2a: Hypocrisy perception plays a mediating role in the mechanism through which “Versailles Literature” language style affects consumers’ negative attitudes towards green consumption behaviors.
H2b: Bragging perception plays a mediating role in the mechanism through which “Versailles Literature” language style affects consumers’ negative attitudes towards posters’ green consumption behaviors.
Comment 3: An overview of the experiments in Section 4 will prepare readers for the coming experiments.
Response 3:
Thank you for your suggestions regarding the experimental overview in Section 4. We have made modifications to the experimental overview section in Section 4. In order to help readers better understand the upcoming experiments, we have added detailed experimental backgrounds, objectives, and design processes in this section, and briefly introduced the specific objectives of each experiment. Therefore, readers can more easily understand the background and purpose of the experiment, and smoothly follow up on the following experimental sections. We believe that these modifications will help improve the readability and structure of the manuscript. Thank you for your valuable feedback, and we look forward to your further feedback. The detailed revised draft is on page 9-10, and the excerpt is as follows:
The purpose of this study was to explore the impact of Versailles literature on green consumption behavior and its mechanism. The main effect, mediating effect, and moderating effect of the “Versailles Literature” language style on green consumption behavior were verified using three experiments: In Experiment 1, a one-way intergroup design (Language Style: Versailles Literature vs. non-Versailles Literature) was used to verify the direct impact of “Versailles Literature” language style on green consumption behavior attitude and to explore its mediating mechanism. Study 1 showed that “Versailles Literature” language style is more likely to trigger negative consumer attitudes towards green consumption behavior than “non-Versailles Literature” language style. Bragging perception and hypocrisy perception play a mediating role in this mechanism. Experiment 2 further introduced the strength of social ties as a moderating variable and used a two-factor design of 2 (language style) x 2 (strength of social ties: stranger vs. acquaintance) to investigate the moderating effect of strength of social ties on “Versailles Literature” language style on green consumption behavior. Experiment 3 strengthened the moderating effect of the strength of social ties through a platform-differentiated design and further enhanced the moderating effect of the strength of social ties. Studies 2 and 3 showed that the strength of social ties plays an important moderating role in this process; for example, when consumers see “Versailles literature” posts published by strangers, they perceive a stronger sense of bragging and hypocrisy, leading to negative attitudes towards the green consumption behavior of posters.
Table 2. Overview of the study.
|
Experiment |
|
Purpose |
Method |
Sample status |
|
Experiment 1 |
Main effect, Mediating effect |
To verify the direct impact of “Versailles Literature” language style on green consumption behavior and its mediating mechanism |
Used a one-way intergroup design (language Style: Versailles Literature vs. non-Versailles Literature) |
N=60 35% male |
|
Experiment 2 |
Moderating effect |
Verify the moderating effect of the strength of social ties |
used a two-factor design of 2 (language style: Versailles Literature vs non-Versailles Literature) x 2 (strength of social ties: stranger vs. acquaintance) |
N=120 28.3% male |
|
Experiment 3 |
Moderating effect |
Platform differentiation design is used to control the strength of social ties and enhance the moderating effect of the strength of social ties |
used a two-factor design of 2 (language style: Versailles Literature vs. non-Versailles Literature) x 2 (strength of social ties: stranger vs. acquaintance) |
N=124 25.8% male |
Comment 4: Regarding the experiments, the authors should provide more details regarding data collection (e.g. participant recruitment; experiment tool/platform), data processing and data analysis.
Response 4:
Thank you for your valuable comments. Based on your suggestions, in the revised version, we have provided a detailed description of the participant recruitment process, including recruitment channels, screening criteria, and sample representativeness. We have added detailed information about participant recruitment, experimental tools and platforms, data processing, and data analysis in the article. Specifically, we have supplemented the recruitment process for participants and provided a detailed description of the usage of the experimental platform. We hope that these modifications can help further clarify the design and analysis process of the experiment, and enhance the transparency and credibility of the manuscript. Thank you again for your valuable suggestion. The detailed revised draft is on page 11 line 485-505, and the excerpt is as follows:
Before recruiting participants, we used G* power software for efficacy analysis to ensure a statistically sufficient sample size. Based on the recommendations of Cohen (1988), the equivalent stress (d=0.80), the significance level alpha=0.05, and the statistical effi-cacy 1-beta=0.85 were set, and the minimum number of participants in each group was calculated to be 30, with a total sample size of 60. In this study, we chose college stu-dents as the research object, considering the characteristics of college students in terms of cultural background, socioeconomic status, and age structure. College students often have strong social media skills and are highly sensitive to popular online culture, mak-ing them an ideal group in which to study digital interactions and the impact of social media. This study issued a recruitment announcement through the Student Affairs Office of a university and invited college students to participate in the experiment via random sampling. The recruitment announcement detailed the research purpose, par-ticipation requirements, and reward measures for participants. A total of 60 students were recruited and divided into two groups to participate in the experiment. All par-ticipants were students who had not been exposed to the experiment, and the partici-pants did not receive relevant training or pre-experimental materials before the ex-periment. The 60 participants were randomly assigned to the two experimental groups using the random number generation function in the SPSS software. All participants signed informed consent forms, and the study was reviewed by the university’s ethics committee (IRB-jxnu-b-20241101, Nov. 1, 2024) and conducted in accordance with the Helsinki Declaration. No personal identification information was collected.
Comment 5: In the design of Exp 2, how was the factor of Social ties manipulated?
Response 5:
Thank you for providing valuable feedback on the design of Experiment 2. Upon re-examination, we realized that we had indeed accidentally overlooked this important factor during the design process. This is indeed our negligence, as we did not clearly explain in the article how we manipulated the closeness of our relationship. Due to the detailed description of key variables in the experimental process section, to avoid redundancy, we only mentioned the guidance in the Materials section and did not repeat it in the Methods section. The original text (Chinese version) provides complete guidance in the supplementary materials, but this part may have been accidentally omitted in the English translation. We apologize for this. Now, we have provided additional clarification, and in the revised manuscript, we will elaborate on how Experiment 2 directly manipulated the closeness of relationships through the use of introductory words in situational materials. We believe that these modifications will make the experimental design more comprehensive, and enhance the depth and accuracy of the research, and we truly appreciate your valuable feedback. The detailed revised draft is on page 14 line 608-614, and the excerpt is as follows:
In order to manipulate the strength of social ties, this study added the following guidance at the beginning of the experimental material: "please imagine the following situation: your good friend Li Ming, you have known each other for more than two years, you have a good relationship, often eat and study together (a fellow student you don’t know, met once and knew his name is Li Ming, but there is no communication), and he (she) posted the following in the circle of friends. Please read this circle of friends news carefully.”
Comment 6: Too many tables are presented in the article. The authors may consider using some figures to increase the readability. Some tables can be included in supplementary materials.
Response 6:
Thank you for your suggestions. We understand that too many tables may affect the readability of the article, so we have streamlined the tables and considered moving some detailed table data to the Appendix. However, some tables contain detailed statistical data that is crucial for understanding the core findings of our research, and these contents may not be effectively presented in the charts. Thus, I kept these tables to help readers better understand the rigor of our analysis and the accuracy of the data. We believe that, although the article contains a certain number of tables, they are an essential component for understanding our research results, and through proper formatting and annotations, we have ensured that these tables do not interfere with the fluency and readability of the article.
Comment 7: Conclusion should come after discussion.
Response 7:
Thank you for your careful review and valuable suggestions on my paper. Regarding the issue you mentioned that the Conclusion should be drawn after the Discussion, in the current version, due to a structural oversight, the Conclusion section was presented earlier, which may have affected the logical fluency of the paper. In the revised manuscript, I moved the Conclusion section to after the Discussion to ensure a clearer and more cohesive logical relationship between the research findings and the Discussion. The Discussion section further deepens the interpretation and analysis of the research results, providing a more comprehensive theoretical and empirical basis for the Conclusion. I believe that such adjustments help improve the overall structure and readability of the paper. Thank you again for your valuable suggestions. I have carefully revised the paper to make it more logical and academic.
Comment 8: The authors should connect their findings to the theories and studies discussed earlier in the paper and provide a clearer interpretation of the results. The current discussion section is underdeveloped and lacks depth.
Response 8:
Thank you for your valuable feedback on this study. I fully agree with your point of view that the in-depth analysis of the Discussion section is indeed a part that needed further strengthening in this article. In response to your feedback, I have re-examined and strengthened the structure of the Discussion section, especially by better integrating the research results with the theoretical framework discussed earlier and existing research. In the process of revision, in order to further explain the connection between our research results and related theories, we reviewed the relevant studies on the language style and greenwashing behavior of “Versailles Literature”, and clearly pointed out how this study provides new perspectives in existing literature. We also fully discuss its theoretical significance in the field of green consumption behavior. We hope that, through these improvements, the research results can be made clearer, further reflecting their academic contributions and practical significance. Thank you again for your valuable suggestions. I will do my best to improve and look forward to your further guidance. The detailed revised draft is on page 22, and the excerpt is as follows:
This study extensively explored the impact of language style (Versailles Literature and non-Versailles Literature) on consumers’ attitudes towards the green consumption behavior of posters based on the theory of self-presentation. It enriches the application and understanding of the “Versailles Literature” language style in the field of green consumption behavior. In the context of green consumption, “Versailles Literature” language style is a rhetorical means of irony, exaggeration, contrast, and so on, which disguises its own achievements in green consumption behavior through superficial humility, complaint, or self-mockery. Previous studies have focused on social psychol-ogy, brand marketing, tourism marketing, and corporate investment (Sezer et al., 2018; Grant et al., 2018; Chen et al., 2020; Feng et al., 2023; Yan et al., 2024). This study ex-plores the impact mechanism of “Versailles Literature” language style on green con-sumption behavior attitude on social media and highlights the unique role of language style in shaping consumer attitudes. Compared with “non-Versailles Literature”, “Ver-sailles Literature” language style is regarded as a hypocritical self-display, thus reduc-ing the audience’s positive attitude towards the green consumption behavior of the poster. This expands the research on self-display strategies on social media and the “Versailles Literature” language style in the field of green consumption (Kim, 2011), and provides inspiration for communication strategies utilized by enterprises and marketers when promoting green consumption.
Secondly, this study reveals the internal mechanism through which the “Versailles Literature” language style affects individuals' attitudes towards green consumption be-havior. Existing studies have explored the single, chain, and parallel mediators of the “Versailles Literature” language style, such as perceived sincerity, reviewer favorability, benign jealousy, amusement, annoyance, ostentation, and sincerity (Sezer et al., 2018; Chen et al., 2020; Paramita & Septiento, 2021; Ma, 2023). Unlike previous studies on cognitive or emotional mediators, this study explores the mediating roles of bragging and hypocrisy perceptions on green consumption, a specific situation with moral sensi-tivity. Compared with “non-Versailles Literature” language style, “Versailles Literature” language style increases the audience's perception of bragging and hypocrisy by the poster, thus reducing the audience’s positive attitude towards the poster’s green con-sumption behavior. The study echoes the central idea of the importance of motivation perception in self-presentation theory. Elucidating the indirect influence of these per-ceptions on attitudes toward green consumption behavior offers a new perspective on how consumers form negative evaluations of consumption behaviors in social interac-tions.
Unlike traditional studies on green consumption behavior that focus primarily on consumers’ intrinsic motivations and socio-cultural backgrounds, and traditional green bleaching behavior research that focuses mainly on the impact of false environmental protection publicity strategies at the enterprise level on consumer cognition and pur-chase behavior (Luo et al., 2017; Shang et al., 2022; Čapienė et al., 2022), this study ex-plores the psychological mechanism of the “green flaunting” phenomenon at the indi-vidual level from the perspective of individual consumer language expression, providing an important supplement to studies on green bleaching behavior from the micro-individual perspective. Combined with the perception theory in social psychol-ogy, this study discusses the impact of language style and social relationships on con-sumer attitudes and reveals the multi-dimensional impact of the complex interweaving of language and social interaction on the formation of green consumption attitudes. Through these theoretical insights, the study provides additional avenues for future research on green consumption behavior and fosters interdisciplinary integration be-tween consumer behavior and social psychology.
Comment 9: Minor issues:
The authors are recommended to pay attention to the formatting. For example, the references should appear before the full stop on Lines 58 and 63.
Please proofread the article carefully. Lines 241, 246, 267, 272 (and more): a space is needed before “language”.
The formatting of the reference list should be consistent. For example, References 28 and 39 provide full names of authors while others only include initials for given names.
The authors may revise some long sentences to increase the readability of the article. For example, Lines 251-254 is difficult to comprehend.
Please do not Chinese punctuation marks. For example, brackets from Line 417.
Since the same materials were used in the formal experiment, there’s no point restating the two posts on Lines 400-412.
Section 4.1.2: Since there are several statements, the authors may consider listing them in a table.
Table 1: No need to include the Cumulative Percentage (%)
Response 9:
We thank the reviewers for their careful examination and valuable comments. We are very sorry for our carelessness and mistakes, and thank you very much for your reminder. According to your comments, we have made detailed changes for each issue, including correcting the formatting and references, simplifying long sentences, correcting punctuation, deleting duplication, adjusting tables, and simplifying Table 1. All changes have been made as required, and we believe that these changes have greatly improved the readability and format consistency of the article.
Reviewer 2 Report (New Reviewer)
Comments and Suggestions for Authors
The manuscript presents a timely and original investigation.
The study is grounded in self-presentation theory and offers meaningful insights by incorporating both mediating (bragging and hypocrisy perception) and moderating (strength of social ties) variables. The use of a multi-experiment design adds rigor and credibility to the findings. Overall, the topic is relevant, the methodology is sound, and the statistical analyses are thorough and appropriately applied.
However, the manuscript would benefit from several improvements in terms of clarity, structure, and precision of language. First, while the theoretical background is comprehensive, the manuscript would be strengthened by a clearer and more succinct articulation of the research gap. For example, lines 69–72 could be revised to explicitly state how the study uniquely contributes to both the green marketing and language style literature. Consider streamlining the introduction to reduce repetition and enhance focus.
Second, the writing style, while generally understandable, contains a number of syntactic and grammatical issues that may hinder comprehension. Examples include awkward constructions such as “which only cost me 500 yuan” (line 405), and unnecessary tautologies like “individuals who... post their behaviors on social media platforms.” A thorough proofreading by a native or proficient English speaker is recommended to improve the fluency and professionalism of the manuscript.
Third, while the results are statistically sound, the presentation of findings would be improved by enhancing the visual clarity of tables and figures. In particular, Figure 1 (line 357) lacks adequate labeling and explanation, making it difficult for readers to quickly understand the proposed model. Adding concise captions that interpret key relationships would aid comprehension.
Moreover, the discussion section could better contextualize the findings within existing literature. For example, lines 495–606 discuss Experiment 2 results but do not sufficiently compare these with prior studies on self-presentation or greenwashing effects. A deeper theoretical integration in the discussion would increase the scholarly contribution of the work.
Finally, the limitations section (line 90 onward) could be expanded to explicitly mention potential biases from student-only samples, cultural context (Chinese participants), and artificiality of experimental stimuli. This would demonstrate critical reflection and openness for future research improvements.
Author Response
Thank you for your valuable comments. We have made significant changes to address each of your concerns and enhance the quality of the manuscript.
Comment 1: However, the manuscript would benefit from several improvements in terms of clarity, structure, and precision of language. First, while the theoretical background is comprehensive, the manuscript would be strengthened by a clearer and more succinct articulation of the research gap. For example, lines 69–72 could be revised to explicitly state how the study uniquely contributes to both the green marketing and language style literature. Consider streamlining the introduction to reduce repetition and enhance focus.
Response 1:
We have made revisions in lines 69-72 to more clearly address the gaps in existing research, particularly pointing out the research gap of “Versailles literature” as an emerging language style in the context of green consumption. We further emphasized the unique contribution of this study in the fields of green marketing and language style. We understand your suggestion to simplify the Introduction to reduce repetition and enhance focus. However, we believe that the content of the Introduction is crucial for providing a comprehensive background and constructing a research framework, especially to help readers understand the background and research objectives of this study. We believe that the structure and content of the Introduction do not exceed the expected length, and each part of the content is of great significance in clarifying the research question and its background. We will optimize the language, simplify redundant parts, and strengthen the structure and logical coherence. The detailed revised draft is on page 2 line 71-78.
Comment 2: Second, the writing style, while generally understandable, contains a number of syntactic and grammatical issues that may hinder comprehension. Examples include awkward constructions such as “which only cost me 500 yuan” (line 405), and unnecessary tautologies like “individuals who... post their behaviors on social media platforms.” A thorough proofreading by a native or proficient English speaker is recommended to improve the fluency and professionalism of the manuscript.
Response 2:
Thank you for providing valuable feedback on the writing style and grammar issues. Firstly, regarding the issue of “awkward structures” and unnecessary synonyms you mentioned, we have reconstructed the relevant sentences to make them more concise and clear. For example, “This only cost me 500 yuan” has been changed to “I only spent 500 yuan”, and corresponding modifications have been made to other similar issues. Secondly, we are aware of some grammar and syntax issues that may have existed in the writing. In order to further improve the fluency and professionalism of the article, we submitted the manuscript to native English-speaking experts for proofreading to ensure that the article meets standard English writing norms and to enhance overall readability. We believe that these improvements will greatly enhance the readability and academic quality of the manuscript. Thank you again for your review and suggestions. We look forward to your further feedback.
Comment 3: Third, while the results are statistically sound, the presentation of findings would be improved by enhancing the visual clarity of tables and figures. In particular, Figure 1 (line 357) lacks adequate labeling and explanation, making it difficult for readers to quickly understand the proposed model. Adding concise captions that interpret key relationships would aid comprehension.
Response 3:
Thank you for pointing out the issue of insufficient labeling and explanation in Figure 1. We have added concise captions based on your suggestion to explain the key relationships in the diagram. In addition, we have added clear labels in the figure and highlighted the main causal relationships and moderating effects in the model. We believe that these modifications can help readers better understand the model structure. The detailed revised draft is on page 9 line 409-422.
Comment 4: Moreover, the discussion section could better contextualize the findings within existing literature. For example, lines 495–606 discuss Experiment 2 results but do not sufficiently compare these with prior studies on self-presentation or greenwashing effects. A deeper theoretical integration in the discussion would increase the scholarly contribution of the work.
Response 4:
Thank you for your valuable feedback on this study. We fully understand your opinion, and regarding the suggestion to place the findings from existing literature in the background in the Discussion section, we have made improvements accordingly. In the revised manuscript, we chose to compare and discuss the language style, self-presentation, and greenwashing behavior related to “Versailles literature” in the Discussion section, rather than conducting in-depth comparisons in the Experimental Results section. This is because we believe that placing these theoretical connections in the Theoretical Contribution section of the paper can present the innovation of this study more systematically and coherently, avoid repetitive discussions in the Experimental Results section, maintain the logical flow of the entire paper, and enhance the logical and academic value of the paper. Thank you again for your valuable feedback. We will further strengthen the Discussion section based on your suggestions to ensure that this study can make more in-depth and academic contributions at both the theoretical and empirical research levels. The detailed revised draft is on page 22, and the excerpt is as follows:
This study extensively explored the impact of language style (Versailles Literature and non-Versailles Literature) on consumers’ attitudes towards the green consumption behavior of posters based on the theory of self-presentation. It enriches the application and understanding of the “Versailles Literature” language style in the field of green consumption behavior. In the context of green consumption, “Versailles Literature” language style is a rhetorical means of irony, exaggeration, contrast, and so on, which disguises its own achievements in green consumption behavior through superficial humility, complaint, or self-mockery. Previous studies have focused on social psychol-ogy, brand marketing, tourism marketing, and corporate investment (Sezer et al., 2018; Grant et al., 2018; Chen et al., 2020; Feng et al., 2023; Yan et al., 2024). This study ex-plores the impact mechanism of “Versailles Literature” language style on green con-sumption behavior attitude on social media and highlights the unique role of language style in shaping consumer attitudes. Compared with “non-Versailles Literature”, “Ver-sailles Literature” language style is regarded as a hypocritical self-display, thus reduc-ing the audience’s positive attitude towards the green consumption behavior of the poster. This expands the research on self-display strategies on social media and the “Versailles Literature” language style in the field of green consumption (Kim, 2011), and provides inspiration for communication strategies utilized by enterprises and marketers when promoting green consumption.
Secondly, this study reveals the internal mechanism through which the “Versailles Literature” language style affects individuals' attitudes towards green consumption be-havior. Existing studies have explored the single, chain, and parallel mediators of the “Versailles Literature” language style, such as perceived sincerity, reviewer favorability, benign jealousy, amusement, annoyance, ostentation, and sincerity (Sezer et al., 2018; Chen et al., 2020; Paramita & Septiento, 2021; Ma, 2023). Unlike previous studies on cognitive or emotional mediators, this study explores the mediating roles of bragging and hypocrisy perceptions on green consumption, a specific situation with moral sensi-tivity. Compared with “non-Versailles Literature” language style, “Versailles Literature” language style increases the audience's perception of bragging and hypocrisy by the poster, thus reducing the audience’s positive attitude towards the poster’s green con-sumption behavior. The study echoes the central idea of the importance of motivation perception in self-presentation theory. Elucidating the indirect influence of these per-ceptions on attitudes toward green consumption behavior offers a new perspective on how consumers form negative evaluations of consumption behaviors in social interac-tions.
Unlike traditional studies on green consumption behavior that focus primarily on consumers’ intrinsic motivations and socio-cultural backgrounds, and traditional green bleaching behavior research that focuses mainly on the impact of false environmental protection publicity strategies at the enterprise level on consumer cognition and pur-chase behavior (Luo et al., 2017; Shang et al., 2022; Čapienė et al., 2022), this study ex-plores the psychological mechanism of the “green flaunting” phenomenon at the indi-vidual level from the perspective of individual consumer language expression, providing an important supplement to studies on green bleaching behavior from the micro-individual perspective. Combined with the perception theory in social psychol-ogy, this study discusses the impact of language style and social relationships on con-sumer attitudes and reveals the multi-dimensional impact of the complex interweaving of language and social interaction on the formation of green consumption attitudes. Through these theoretical insights, the study provides additional avenues for future research on green consumption behavior and fosters interdisciplinary integration be-tween consumer behavior and social psychology.
Comment 5: Finally, the limitations section (line 90 onward) could be expanded to explicitly mention potential biases from student-only samples, cultural context (Chinese participants), and artificiality of experimental stimuli. This would demonstrate critical reflection and openness for future research improvements.
Response 5:
Thank you for your valuable suggestion. We fully agree with your suggestion to expand the restricted section. Regarding the limitations of sample size and population composition, we have made corresponding supplements and improvements in the revised draft. In the Limitations section of our research, we further clarified the small sample size and sample population, and indicated that, in future studies, we will expand the sample size to include individuals from more age groups, occupational groups, and other social backgrounds, in order to improve the generalizability and external validity of the research results. We believe that these modifications can more comprehensively reflect the limitations of the research and provide valuable insights and directions for future research. Thank you again for your valuable suggestion. The detailed revised draft is on page 24, and the excerpt is as follows:
Secondly, the participants consisted of a small number of students from a single university in southern China; however, there are significant differences in economic development levels, cultural traditions, and consumption concepts in different regions. Additionally, the participants were between 18 and 25 years old. Consumers in this age group usually have strong social media use ability, high sensitivity to popular online culture, and are in the critical period of consumption concepts and value formation. Consumers of different ages may have significant differences in terms of social media use habits, language understanding ability, and green consumption cognitive levels. Thus, the findings of this study may be more relevant to younger people, with their relevance to other age groups or groups of people from different backgrounds being unclear. Future studies should consider expanding the sample to include individuals of more diverse ages, occupations, and social backgrounds in order to improve the gener-alizability and external validity of the results.
Thirdly, the study focused on only two language styles—“Versailles Literature” and “non-Versailles Literature”. Future research could explore a more diverse range of language styles and investigate their potential impacts on individuals’ attitudes toward the green consumption behaviors of posters. Finally, the findings of this study are lim-ited by the specific design of the selected platform. The study relied on a single Chinese social media platform whose design and functionality may differ significantly from those of other social media platforms, such as Facebook or Instagram, which are global social media platforms. This may limit the generalizability of this study’s conclusions to other digital communication environments. Furthermore, different social media plat-forms may exert varying influences on consumers’ language styles and attitudes. For example, visually oriented platforms (such as Xiaohongshu and Douyin) may have different effects on language style compared with text-based social media platforms (such as Weibo and Zhihu). Future research could examine the moderating role of platform type on the effectiveness of language style on social media marketing.
Reviewer 3 Report (New Reviewer)
Comments and Suggestions for Authors
Dear Authors,
Your topic is interesting in the fields of consumer behavior and digital marketing. However, to enhance the paper's theoretical clarity, analytical transparency, and empirical credibility, several substantive and methodological areas require improvement.
Below, please find my feedback:
- Abstract
To exhibit research transparency, the sample size, methodological approach, and principal findings would enhance the informativeness and rigor of the abstract.
- Introduction
- A more straightforward introduction of the "Versailles Literature" and a more thorough articulation of the research gap and contribution are necessary for the introduction of greenwashing and corporate language strategies.
- It is essential to provide context for the term "Versailles Literature" and to explain its emergence in literary and cultural criticism.
- You must improve this section by informing scholars of the topic's importance in academic discussions, particularly in discourse analysis, environmental communication, and critical management.
- The research objectives and questions should be part of the introduction.
- Should policies, cultural facts, or any other elements be explained in this introduction?
- Theoretical background
- To improve readability and logical flow, the section should exhibit a more unified organizational structure.
- The development of ideas, such as corporate boasting and hypocrisy, appears fragmented.
- It is necessary to have a more robust theoretical foundation by rearranging these subsections. Maybe your arguments should shift from descriptive aspects of language style to their ethical implications.
- Connecting each subsection to the related hypotheses is necessary. Each hypothesis should be supported with at least 150 words.
- Theories have been developed to some extent beyond the earlier debates, which could create an ambiguous understanding of their theoretical justification.
- The coherence of arguments would benefit from making clear how each aspect of language style influences particular expectations.
- To benefit the readers, a comparative table including main language styles, characteristics, impacts, and predicted results would be helpful.
- Hypothesis development
- It is necessary to provide more explanation to support hypotheses 3a and 3b and clarify their implications.
- The moderated mediation model must be improved theoretically.
- The way hypotheses are presented needs to be improved and rearranged to enhance the research framework.
- Methodology
- The manuscript would benefit from a better explanation of the sample's demographic and cultural limitations.
- The methodology would benefit from a better explanation of the randomization and participant recruitment processes.
- A table listing the salient characteristics of each of the three experiments is necessary to improve clarity.
- Results
- Although Composite Reliability (CR), Average Variance Extracted (AVE), and standardized factor loadings are reported, the evaluation of discriminant validity is lacking.
- There is no correlation matrix between the latent constructs. This limits the capacity to evaluate discriminant validity.
- The results and methods do not apply or explain the Fornell-Larcker criterion, despite its brief mention.
- The Heterotrait-Monotrait Ratio (HTMT), commonly acknowledged as a reliable indicator of discriminant validity, is neither computed nor discussed.
- Maximum Shared Variance (MSV) and MaxR(H) are not reported.
- Discussion
- A thorough examination of the psychological processes underlying the observed effects would enhance the conversation.
- The manuscript should acknowledge the limitations of the suggested model and more directly examine possible alternative explanations.
- The study's contribution must be enhanced.
- Theoretical contributions
- In many places, the manuscript currently overstates the novelty of its contribution.
- The definition of "Versailles Literature" is not entirely clear.
- Managerial implications
- Extending the manuscript's implications beyond the Chinese platform context would strengthen it.
- Limitations
- The sample's demographic homogeneity raises questions about generalizability, particularly regarding age, education, and cultural background.
- The applicability of the findings to other digital communication environments is limited by their platform-specific design, which is tied to a single Chinese social media site.
- Language and clarity
- Professional language editing is necessary to improve the manuscript's readability and clarity.
- It is necessary to review the paper's terminology carefully to maintain consistency.
Comments on the Quality of English Language
English proofreading is necessary.
Author Response
Thank you very much for your meticulous review of every part of the paper, from the Abstract and Introduction to the Theoretical Background, Experimental Design, Data Analysis and Conclusion, and for providing corresponding suggestions. The responses are as follows one by one:
Comment 1: Abstract
To exhibit research transparency, the sample size, methodological approach, and principal findings would enhance the informativeness and rigor of the abstract.
Response 1:
Thank you for your valuable suggestions for the Summary section. We fully agree with you that providing a more detailed description of sample sizes, methods, and major findings will effectively improve the transparency and rigor of the study. In the revised summary, we have made the following changes based on your comments. The summary explicitly mentioned the sample size, research methods, and data analysis software used in the study. Thank you again for your positive comments and valuable suggestions to improve the quality of our manuscripts. The detailed revised draft is on page 1, and the excerpt is as follows:
The dissemination and practice of green consumption behavior is an important issue in promoting sustainable development. With the advent of the digital age, social media platforms have become an important channel for promoting green consumption. The expression of language style has become an increasingly important factor influencing consumer attitudes. From the perspective of consumer perception, this study used three situational simulation experiments (n total=304) to explore the mechanism by which the “Versailles Literature” language style impacts the feelings and behaviors of audiences of the green consumption behavior of the poster, and to examine the mediating roles of ostentation perception and hypocrisy perception. Data analysis was conducted using SPSS. The research findings showed that, compared with “non-Versailles Literature”, this style significantly reduces positive attitudes toward green consumption while increasing percep-tions of bragging and hypocrisy. Furthermore, the strength of social ties between the consumer and the poster plays a moderating role in the effect of language style; specifically, when posts come from strangers, consumers perceive a stronger sense of bragging and hypocrisy. The research results will provide practical guidance for individuals and enterprises to effectively promote the concept of green consumption on social media, helping enterprises avoid the negative reactions brought about by conspicuous green consumption behaviors and exaggerated or false promotion of environmental behaviors, such as “greenwashing”.
Comment 2: Introduction
A more straightforward introduction of the "Versailles Literature" and a more thorough articulation of the research gap and contribution are necessary for the introduction of greenwashing and corporate language strategies.
It is essential to provide context for the term "Versailles Literature" and to explain its emergence in literary and cultural criticism.
You must improve this section by informing scholars of the topic's importance in academic discussions, particularly in discourse analysis, environmental communication, and critical management.
The research objectives and questions should be part of the introduction.
Should policies, cultural facts, or any other elements be explained in this introduction?
Response 2:
Thank you for your valuable suggestions on the Introduction section. We have made the following modifications to the Introduction based on your feedback. In the Introduction section, we briefly explained the first mention of “Versailles literature”, briefly mentioned the cultural and policy background related to green consumption, and clarified the research question and limitations of existing research. We also clearly stated the research direction of this article, so that readers can quickly understand the core content of the research. The detailed revised draft is on page 2-3.
Comment 3: heoretical background
To improve readability and logical flow, the section should exhibit a more unified organizational structure.
The development of ideas, such as corporate boasting and hypocrisy, appears fragmented.
It is necessary to have a more robust theoretical foundation by rearranging these subsections. Maybe your arguments should shift from descriptive aspects of language style to their ethical implications.
Connecting each subsection to the related hypotheses is necessary. Each hypothesis should be supported with at least 150 words.
Theories have been developed to some extent beyond the earlier debates, which could create an ambiguous understanding of their theoretical justification.
The coherence of arguments would benefit from making clear how each aspect of language style influences particular expectations.
To benefit the readers, a comparative table including main language styles, characteristics, impacts, and predicted results would be helpful.
Response 3:
Thank you very much for your valuable feedback and constructive suggestions on the theoretical part of this article. Your opinion accurately points out the shortcomings of the current theoretical framework, especially in terms of logical coherence, theoretical integration, and readability. Regarding the discourse on the development of ideas such as “corporate boasting and hypocrisy”, you mentioned that these contents currently appear fragmented, which is indeed due to my failure to coordinate and lay them out well in the writing process. Therefore, in order to improve readability and logical flow, we have comprehensively reconstructed the perception of hypocrisy in the theoretical background, established a clear conceptual development context, and provided a more unified and coherent explanation of the issue of hypocrisy in corporate environmental protection propaganda. Your feedback will significantly enhance the theoretical contribution and academic value of this article. We deeply appreciate it and will conscientiously implement these suggestions in the revised manuscript.
Secondly, we have revised the Theoretical Background section by adopting a more unified and logically clear framework, shifting the focus from the descriptive features of language style to its moral meaning and psychological mechanisms. Finally, regarding your constructive suggestion of adding a comparison table, it does help readers to have a more intuitive understanding of the differences between different language styles. We deeply understand your good intention of improving the readability and information presentation efficiency of the article. In the revised manuscript, we have conducted a detailed textual comparison between the “Versailles Literature” language style and traditional expressions such as “direct bragging” and “virtue signaling” in the Theoretical Background section, and systematically compared and analyzed them from dimensions such as expression strategies, rhetorical devices, and psychological mechanisms. Thank you again for your valuable feedback. I will further revise the paper based on your feedback to make it more complete. The detailed revised draft is on page 3-5.
Table 1. Comparison of the characteristics and influence of different language styles.
|
Characteristic dimension |
Versailles Literature |
Direct bragging |
Virtue signaling |
|
Apply perspective |
Third-person perspective |
First-person perspective |
collective identity expression |
|
Comparison strategy |
reverse contrast, desire to rise first and restrain first |
positive comparison or independent statement |
contrast with unethical behavior |
|
Language characteristics |
ostensibly humble, ostensibly ostentatious |
explicit self-praise |
values Manifesto |
|
Social influence |
arouse audience disgust and reduce social support |
enhance impression and trigger jealousy |
enhance group identity and may trigger moral licensing effect |
|
Related literature |
Scopelliti et al. (2015);Sezer et al. (2018) |
Sedikides & Gregg (2008);Grant et al.,2018 |
Jordan & Monin (2008);Konuk et al.,2024 |
Comment 4: Hypothesis development
It is necessary to provide more explanation to support hypotheses 3a and 3b and clarify their implications.
The moderated mediation model must be improved theoretically.
The way hypotheses are presented needs to be improved and rearranged to enhance the research framework.
Response 4:
Thank you for your meticulous review and valuable feedback on my paper. Your feedback accurately points out the key shortcomings in the current research framework, and these suggestions are of great value in enhancing the theoretical contribution of this article. Based on your feedback, first of all, we will strengthen the role of social cognitive theory and psychological distance theory in explaining the mechanism of relationship strength regulation, provide a more detailed theoretical derivation process for Hypotheses 3a and 3b, and clarify how strong relationships affect the perception of hypocrisy and bragging through different paths. After rethinking the presentation of hypotheses, we have adopted a hierarchical hypothesis structure to more clearly express the research logic. By strengthening the intrinsic connection between hypotheses and research frameworks, we strive to make the entire paper more systematic and logical. Your professional guidance will significantly enhance the academic quality and theoretical depth of this research, and we deeply appreciate it. The detailed revised draft is on page 7-8.
Comment 5: Methodology
The manuscript would benefit from a better explanation of the sample's demographic and cultural limitations.
The methodology would benefit from a better explanation of the randomization and participant recruitment processes.
A table listing the salient characteristics of each of the three experiments is necessary to improve clarity.
Response 5:
Thank you for your valuable feedback on our manuscript. We will make corresponding improvements in the revised version based on your suggestions. Firstly, in the Experimental Design section, we further elaborated on the characteristics of the college student population in terms of cultural background, socio-economic status, and age structure. In the Research Limitations section, we will provide a more detailed explanation of the cultural and demographic limitations of the research sample, so that readers can have a clearer understanding of the impact these factors may have on the generalizability of the research results. Secondly, regarding the randomization and participant recruitment process, we have added a research overview in the Research Design section and further elaborated on the random allocation process and recruitment steps for participants to ensure the transparency and reliability of the research methods. Based on your suggestion, we have added a table that succinctly lists the main characteristics of the three experiments to improve the clarity and readability of the article. Thank you again for your review and suggestions. We believe that these revisions will make the manuscript more complete. The detailed revised draft is on page 9-18, and the excerpt is as follows:
The purpose of this study was to explore the impact of Versailles literature on green consumption behavior and its mechanism. The main effect, mediating effect, and moderating effect of the “Versailles Literature” language style on green consumption behavior were verified using three experiments: In Experiment 1, a one-way intergroup design (Language Style: Versailles Literature vs. non-Versailles Literature) was used to verify the direct impact of “Versailles Literature” language style on green consumption behavior attitude and to explore its mediating mechanism. Study 1 showed that “Ver-sailles Literature” language style is more likely to trigger negative consumer attitudes towards green consumption behavior than “non-Versailles Literature” language style. Bragging perception and hypocrisy perception play a mediating role in this mechanism. Experiment 2 further introduced the strength of social ties as a moderating variable and used a two-factor design of 2 (language style) x 2 (strength of social ties: stranger vs. acquaintance) to investigate the moderating effect of strength of social ties on “Versailles Literature” language style on green consumption behavior. Experiment 3 strengthened the moderating effect of the strength of social ties through a plat-form-differentiated design and further enhanced the moderating effect of the strength of social ties. Studies 2 and 3 showed that the strength of social ties plays an important moderating role in this process; for example, when consumers see “Versailles literature” posts published by strangers, they perceive a stronger sense of bragging and hypocrisy, leading to negative attitudes towards the green consumption behavior of posters.
Table 2. Overview of the study.
|
Experiment |
|
Purpose |
Method |
Sample status |
|
Experiment 1 |
Main effect, Mediating effect |
To verify the direct impact of “Versailles Literature” language style on green consumption behavior and its mediating mechanism |
Used a one-way intergroup design (language Style: Versailles Literature vs. non-Versailles Literature) |
N=60 35% male |
|
Experiment 2 |
Moderating effect |
Verify the moderating effect of the strength of social ties |
used a two-factor design of 2 (language style: Versailles Literature vs non-Versailles Literature) x 2 (strength of social ties: stranger vs. acquaintance) |
N=120 28.3% male |
|
Experiment 3 |
Moderating effect |
Platform differentiation design is used to control the strength of social ties and enhance the moderating effect of the strength of social ties |
used a two-factor design of 2 (language style: Versailles Literature vs. non-Versailles Literature) x 2 (strength of social ties: stranger vs. acquaintance) |
N=124 25.8% male |
Comment 6: Results
Although Composite Reliability (CR), Average Variance Extracted (AVE), and standardized factor loadings are reported, the evaluation of discriminant validity is lacking.
There is no correlation matrix between the latent constructs. This limits the capacity to evaluate discriminant validity.
The results and methods do not apply or explain the Fornell-Larcker criterion, despite its brief mention.
The Heterotrait-Monotrait Ratio (HTMT), commonly acknowledged as a reliable indicator of discriminant validity, is neither computed nor discussed.
Maximum Shared Variance (MSV) and MaxR(H) are not reported.
Response 6:
Thank you very much for your valuable feedback on the discriminant validity evaluation in our research. Your comments have made us realize that there are significant omissions in the evaluation of construct validity, and these opinions have important guiding significance for improving the quality of research. Although we reported aggregation validity indicators such as composite reliability (CR), average variance extraction (AVE), and standardized factor loading, as you pointed out, we were unable to provide a complete discriminant validity evaluation framework. This negligence reflects the limitations of our knowledge in scale validity assessment methods. In the revised manuscript, we have added a complete latent concept correlation matrix, with diagonal lines displaying the square root values of AVE for each concept, to facilitate the intuitive judgment of the Fornell Larcker criterion. From the data in the table, it can be seen that the square root of AVE for each variable in Experiments 1 and 2 is greater than the correlation coefficient between this variable and other variables, and the HTMT values between all variables are below the threshold of 0.90, indicating that the scale in this study has good discriminant validity. These results collectively indicate that our measurement model has good discriminant validity. However, due to space constraints, we plan to briefly mention the relevant results in the article, including the specific correlation matrix HTMT. The calculation results are included in the Appendix. This will help avoid lengthy explanations while ensuring the conciseness of the research and clear expression of the core content. We also greatly appreciate you pointing out the lack of evaluation for discriminant validity and introducing multiple reliable evaluation metrics, such as the Fornell Larcker criterion, HTMT, MSV, and MaxR (H). These suggestions have taught us many new methods and theories, which will have a long-term positive impact on our future research. Thank you again for your valuable feedback.
Experiment 1 Validity Analysis Validity Analysis
|
variables |
CR |
AVE |
BP |
HP |
LS |
|
BP |
0.948 |
0.860 |
0.927 |
|
|
|
HP |
0.963 |
0.897 |
0.887*** |
0.948 |
|
|
LS |
0.944 |
0.851 |
-0.662*** |
-0.641*** |
0.923 |
Experiment 1 HTMT Analysis
|
variables |
BP |
HP |
LS |
|
BP |
|
|
|
|
HP |
0.892 |
|
|
|
LS |
0.655 |
0.631 |
|
Experiment 2 Validity Analysis Validity Analysis
|
variables |
CR |
AVE |
BP |
HP |
LS |
|
BP |
0.923 |
0.800 |
0.894 |
|
|
|
HP |
0.951 |
0.866 |
0.884*** |
0.931 |
|
|
LS |
0.955 |
0.876 |
-0.833*** |
-0.894*** |
0.935 |
Experiment 2 HTMT Analysis
|
variables |
BP |
HP |
LS |
|
BP |
|
|
|
|
HP |
0.884 |
|
|
|
LS |
0.833 |
0.893 |
|
Henseler, J., C. M. Ringle, and M. Sarstedt (2015). A New Criterion for Assessing Discriminant Validity in Variance-based Structural Equation Modeling, Journal of the Academy of Marketing Science, 43 (1), 115-135.
Comment 7: Discussion
A thorough examination of the psychological processes underlying the observed effects would enhance the conversation.
The manuscript should acknowledge the limitations of the suggested model and more directly examine possible alternative explanations.
The study's contribution must be enhanced.
Response 7:
Thank you for your valuable suggestions on our research. We fully agree with your suggestion to strengthen the examination of the psychological processes behind the observed effects. During the revision process, we delved deeper into the psychological mechanisms that influence these effects and further refined the theoretical explanation of the underlying processes. We also attach great importance to the research limitations and alternative explanations you mentioned. We more clearly point out these limitations in the Research Limitations section of the revised manuscript to ensure the rigor and comprehensiveness of the research. Finally, regarding the contribution of strengthening research, we further enhanced the Discussion in this section. In the process of revision, in order to further explain the connection between our research results and related theories, we reviewed the relevant studies on the language style and greenwashing behavior of “Versailles literature”, and clearly pointed out how this study provides new perspectives in existing literature. We also fully discuss its theoretical significance in the field of green consumption behavior to ensure that our research brings more valuable insights to the academic field. Thank you for your valuable feedback. We will make every effort to improve our research. The detailed revised draft is on page 22-24.
Comment 8: Theoretical contributions
In many places, the manuscript currently overstates the novelty of its contribution.
The definition of "Versailles Literature" is not entirely clear.
Response 8:
Thank you very much for your valuable feedback on our manuscript. We have made corresponding modifications based on your suggestions regarding the issue you mentioned that the manuscript exaggerates the novelty of its contributions in many places. Specifically, we have made adjustments to the description of research contributions in the manuscript, avoiding absolute language and instead expressing further exploration and contributions based on existing literature with greater caution. In addition, regarding the issue of unclear definition of “Versailles Literature” that you mentioned, we have provided a clear definition of “Versailles Literature” in the Theoretical Background section of the manuscript based on relevant literature. However, due to our formatting and expression issues, it may have been inconvenient for you to find the definition of “Versailles Literature”. We apologize for any inconvenience this may have caused. For your convenience, we have marked it in blue font in the text. We further clarified the definition of “Versailles Literature” in the Theoretical Contribution section. We believe that these modifications can better demonstrate the actual contribution of this study and hope to further enhance the academic quality of the manuscript. Thank you for your careful review and valuable suggestions. We look forward to your further feedback. The detailed revised draft is on page 22-24.
Comment 9: Managerial implications
Extending the manuscript's implications beyond the Chinese platform context would strengthen it.
Response 9:
Thank you very much for your valuable feedback. As we have clearly pointed out in our manuscript that the research object “Versailles literature” is a unique phenomenon of Chinese internet culture, it has significant cultural specificity. In the Theoretical Background section, we have already indicated the similarity between the “humblebragging” phenomenon on Western social media and “Versailles literature”. Thus, users of Western social platforms may also have similar understandings and emotional reactions to these language styles, which may provide useful references for understanding similar language phenomena in cross-cultural contexts. The detailed revised draft is on page 23 line 930-939.
Comment 10: Limitations
The sample's demographic homogeneity raises questions about generalizability, particularly regarding age, education, and cultural background.
The applicability of the findings to other digital communication environments is limited by their platform-specific design, which is tied to a single Chinese social media site.
Language and clarity
Response 10:
Thank you for your valuable suggestion. We have made corresponding supplements and improvements in the revised manuscript regarding the limitations of sample size and population composition. In the Limitations section of our research, we further clarified the small sample size and sample population, and indicated that, in future studies, we will expand the sample size to include individuals from more age groups, occupational groups, and other social backgrounds, in order to improve the generalizability and external validity of the research results. We further elaborated on the impact of platform-specific design on the applicability of the research conclusions. Due to the reliance of this study on a single Chinese social media platform, the design of the platform may limit the generalizability of the research results to other social media platforms or digital communication environments. Future research will consider the design and functional differences of different platforms and explore their impact on the dissemination of green consumption behavior. We believe that these modifications will help improve the clarity and scientificity of the manuscript. Thank you again for your valuable suggestions. The detailed revised draft is on page 24, and the excerpt is as follows:
Secondly, the participants consisted of a small number of students from a single university in southern China; however, there are significant differences in economic development levels, cultural traditions, and consumption concepts in different regions. Additionally, the participants were between 18 and 25 years old. Consumers in this age group usually have strong social media use ability, high sensitivity to popular online culture, and are in the critical period of consumption concepts and value formation. Consumers of different ages may have significant differences in terms of social media use habits, language understanding ability, and green consumption cognitive levels. Thus, the findings of this study may be more relevant to younger people, with their relevance to other age groups or groups of people from different backgrounds being unclear. Future studies should consider expanding the sample to include individuals of more diverse ages, occupations, and social backgrounds in order to improve the gener-alizability and external validity of the results.
Thirdly, the study focused on only two language styles—“Versailles Literature” and “non-Versailles Literature”. Future research could explore a more diverse range of language styles and investigate their potential impacts on individuals’ attitudes toward the green consumption behaviors of posters. Finally, the findings of this study are lim-ited by the specific design of the selected platform. The study relied on a single Chinese social media platform whose design and functionality may differ significantly from those of other social media platforms, such as Facebook or Instagram, which are global social media platforms. This may limit the generalizability of this study’s conclusions to other digital communication environments. Furthermore, different social media plat-forms may exert varying influences on consumers’ language styles and attitudes. For example, visually oriented platforms (such as Xiaohongshu and Douyin) may have different effects on language style compared with text-based social media platforms (such as Weibo and Zhihu). Future research could examine the moderating role of platform type on the effectiveness of language style on social media marketing.
Comment 11: Professional language editing is necessary to improve the manuscript's readability and clarity. It is necessary to review the paper's terminology carefully to maintain consistency.
Response 11:
Thank you for providing valuable feedback on grammar issues. We fully agree with your viewpoint that professional language editing is crucial for improving the readability and clarity of papers. In order to further improve the fluency and professionalism of the article, we have submitted the manuscript to native English-speaking experts for proofreading to ensure that the article meets standard English writing norms and enhances overall readability. During the editing process, we paid special attention to the consistency of terminology, avoiding the use of different expressions to describe the same concepts, and providing unified definitions and usage of key terms. In addition, we ensured the consistency of professional terminology throughout the entire paper and conducted a terminology review to ensure accuracy and consistency in expression. We believe that these revisions will significantly improve the quality of the manuscript and make it clearer and easier to understand. Thank you for your careful review, and we look forward to your further feedback.
Reviewer 4 Report (New Reviewer)
Comments and Suggestions for Authors
The subject of this study is highly relevant and possesses significant scientific and societal implications. Its importance is particularly emphasized by the limited existing research on the "Versailles Literature" language style in social media and its impact on consumers’ perceptions of green consumption.
The study provides a precise definition of “Versailles Literature” and distinguishes it from non-Versailles content in a way that is easy to replicate. The hypothetical and methodological framework is well developed and coherent. The hypotheses are properly defined, making them testable in practice. All proposed hypotheses (H1–H3b) are supported by statistically significant results.
In addition, the use of the English language is appropriate and understandable to a wider audience. The example posts are well-constructed and align with the theoretical concept being tested.
Furthermore, the study utilized a comprehensive method of literature analysis, systematically reviewing and critically engaging with up-to-date and relevant sources. Such an approach ensured a robust theoretical foundation and contextual relevance for the research.
While acknowledging the evident effort the authors have invested in this research, the quality of the manuscript can be improved through the following revisions.
Discussion section - The discussion section would be strengthened by incorporating further comparisons with relevant studies to contextualize the findings within the existing body of literature.
Research limitations - A notable limitation of the study is the relatively small sample size, comprising only 60 university students. Additionally, the demographic composition of the sample is skewed, with the respondents being younger than 30 years old. This limited and demographically narrow sample restricts the generalizability of the findings to broader populations. It is important that the authors explicitly acknowledge these constraints in the limitations section to provide transparency regarding the scope and applicability of the study’s results.
Overall conclusion - The study explores the influence of Versailles Literature, as a culturally specific and theoretically relevant language style, on perceptions of green behaviour. The research is novel and contributes to both consumer behaviour and communication studies. Conducting three experiments with increasing realism and complexity significantly enhances the validity and generalizability of the findings. The manuscript's writing style and presentation of results are clear and approachable for a broad audience, extending beyond the academic community, which enhances its potential to engage a wide range of readers. By addressing the recommended corrective measures, the authors will significantly enhance the manuscript’s overall quality and academic relevance.
Author Response
Thank you for your detailed review of our manuscript and valuable suggestions. The opinions you put forward have been of great significance for us to further improve the research. We have made corresponding revisions and supplements to the manuscript based on your feedback. The responses are as follows one by one:
Comment 1: Discussion section - The discussion section would be strengthened by incorporating further comparisons with relevant studies to contextualize the findings within the existing body of literature.
Response 1:
Thank you for your valuable suggestions on the Discussion section of our paper. We strongly agree with the suggestion of further comparing research results with relevant studies to place them in the context of existing literature, and recognize that this will further enhance the academic depth and rigor of the paper. In the revised manuscript, we further strengthen this section by reviewing relevant research on the language style and greenwashing behavior of “Versailles literature”, and clearly pointing out how this study provides new perspectives in existing literature. We believe that these modifications will help enhance the theoretical contribution of the paper, enabling our research findings to better integrate and drive the development of existing literature. Thank you again for your valuable feedback. Based on your feedback, we have made more detailed and comprehensive revisions to the Discussion section to enhance the academic value of the paper. The detailed revised draft is on page 22, and the excerpt is as follows:
This study extensively explored the impact of language style (Versailles Literature and non-Versailles Literature) on consumers’ attitudes towards the green consumption behavior of posters based on the theory of self-presentation. It enriches the application and understanding of the “Versailles Literature” language style in the field of green consumption behavior. In the context of green consumption, “Versailles Literature” language style is a rhetorical means of irony, exaggeration, contrast, and so on, which disguises its own achievements in green consumption behavior through superficial humility, complaint, or self-mockery. Previous studies have focused on social psychology, brand marketing, tourism marketing, and corporate investment (Sezer et al., 2018; Grant et al., 2018; Chen et al., 2020; Feng et al., 2023; Yan et al., 2024). This study explores the impact mechanism of “Versailles Literature” language style on green consumption behavior attitude on social media and highlights the unique role of language style in shaping consumer attitudes. Compared with “non-Versailles Literature”, “Versailles Literature” language style is regarded as a hypocritical self-display, thus reducing the audience’s positive attitude towards the green consumption behavior of the poster. This expands the research on self-display strategies on social media and the “Versailles Literature” language style in the field of green consumption (Kim, 2011), and provides inspiration for communication strategies utilized by enterprises and marketers when promoting green consumption.
Secondly, this study reveals the internal mechanism through which the “Versailles Literature” language style affects individuals' attitudes towards green consumption behavior. Existing studies have explored the single, chain, and parallel mediators of the “Versailles Literature” language style, such as perceived sincerity, reviewer favorability, benign jealousy, amusement, annoyance, ostentation, and sincerity (Sezer et al., 2018; Chen et al., 2020; Paramita & Septiento, 2021; Ma, 2023). Unlike previous studies on cognitive or emotional mediators, this study explores the mediating roles of bragging and hypocrisy perceptions on green consumption, a specific situation with moral sensitivity. Compared with “non-Versailles Literature” language style, “Versailles Literature” language style increases the audience's perception of bragging and hypocrisy by the poster, thus reducing the audience’s positive attitude towards the poster’s green consumption behavior. The study echoes the central idea of the importance of motivation perception in self-presentation theory. Elucidating the indirect influence of these perceptions on attitudes toward green consumption behavior offers a new perspective on how consumers form negative evaluations of consumption behaviors in social interactions.
Unlike traditional studies on green consumption behavior that focus primarily on consumers’ intrinsic motivations and socio-cultural backgrounds, and traditional green bleaching behavior research that focuses mainly on the impact of false environmental protection publicity strategies at the enterprise level on consumer cognition and purchase behavior (Luo et al., 2017; Shang et al., 2022; Čapienė et al., 2022), this study explores the psychological mechanism of the “green flaunting” phenomenon at the individual level from the perspective of individual consumer language expression, providing an important supplement to studies on green bleaching behavior from the micro-individual perspective. Combined with the perception theory in social psychology, this study discusses the impact of language style and social relationships on consumer attitudes and reveals the multi-dimensional impact of the complex interweaving of language and social interaction on the formation of green consumption attitudes. Through these theoretical insights, the study provides additional avenues for future research on green consumption behavior and fosters interdisciplinary integration between consumer behavior and social psychology.
Comment 2: Research limitations - A notable limitation of the study is the relatively small sample size, comprising only 60 university students. Additionally, the demographic composition of the sample is skewed, with the respondents being younger than 30 years old. This limited and demographically narrow sample restricts the generalizability of the findings to broader populations. It is important that the authors explicitly acknowledge these constraints in the limitations section to provide transparency regarding the scope and applicability of the study’s results.
Response 2:
Thank you for your valuable suggestion. We have made corresponding supplements and improvements in the revised manuscript regarding the limitations of sample size and population composition. We point out the small sample size and sample population in the Study Limitations section and indicate that, for future studies, we will expand the sample to include more individuals from more age groups, occupational groups, and other social backgrounds in order to increase the generalizability and external validity of the findings. We believe that these modifications will help improve the clarity and scientificity of the manuscript. Thank you again for your valuable suggestions. The detailed revised draft is on page 24, and the excerpt is as follows:
Secondly, the participants consisted of a small number of students from a single university in southern China; however, there are significant differences in economic development levels, cultural traditions, and consumption concepts in different regions. Additionally, the participants were between 18 and 25 years old. Consumers in this age group usually have strong social media use ability, high sensitivity to popular online culture, and are in the critical period of consumption concepts and value formation. Consumers of different ages may have significant differences in terms of social media use habits, language understanding ability, and green consumption cognitive levels. Thus, the findings of this study may be more relevant to younger people, with their relevance to other age groups or groups of people from different backgrounds being unclear. Future studies should consider expanding the sample to include individuals of more diverse ages, occupations, and social backgrounds in order to improve the generalizability and external validity of the results.
Thirdly, the study focused on only two language styles—“Versailles Literature” and “non-Versailles Literature”. Future research could explore a more diverse range of language styles and investigate their potential impacts on individuals’ attitudes toward the green consumption behaviors of posters. Finally, the findings of this study are limited by the specific design of the selected platform. The study relied on a single Chinese social media platform whose design and functionality may differ significantly from those of other social media platforms, such as Facebook or Instagram, which are global social media platforms. This may limit the generalizability of this study’s conclusions to other digital communication environments. Furthermore, different social media platforms may exert varying influences on consumers’ language styles and attitudes. For example, visually oriented platforms (such as Xiaohongshu and Douyin) may have different effects on language style compared with text-based social media platforms (such as Weibo and Zhihu). Future research could examine the moderating role of platform type on the effectiveness of language style on social media marketing.
Round 2
Reviewer 3 Report (New Reviewer)
Comments and Suggestions for Authors
Dear authors,
I want to thank you for all the improvements to your paper. I recognize the efforts and outcome. I believe the paper is now ready for publication.
I wish you the best in this peer review process.
Author Response
Response to Reviewer 3
Thank you very much for the meticulous review work of the reviewers. We are deeply sorry for the incomplete spelling of the words in the picture you pointed out, and we are responsible for this negligence. We have carefully checked and corrected spelling errors in all charts, conducted a comprehensive format and normative review of the entire manuscript as recommended by you, and adjusted the format of punctuation, references, etc. to ensure that all content meets the required standards of the journal. We are keenly aware of the importance of detail and pledge to be more rigorous in future academic writing. Thank you again for your patient and meticulous review work, your professional attitude and responsible spirit have benefited us a lot, and also prompted us to constantly improve the quality of academic papers.
This manuscript is a resubmission of an earlier submission. The following is a list of the peer review reports and author responses from that submission.
Round 1
Reviewer 1 Report
Comments and Suggestions for Authors
The concept of "Versailles literature language" lacks theoretical support in this study. The cited references do not include relevant studies that substantiate the application of this concept in the marketing field. Furthermore, the grouping of diverse literary figures under this concept appears deliberate and may only coincidentally align with the features used to define it. Therefore, it is important to note that there are insufficient metalinguistic elements to support the bragging intention of the post's author. Consequently, multiple interpretations can arise from the text, introducing potential bias based on the questions posed to participants.
The primary issue lies in the construction of the stimuli. The post presented to participants does not provide any contextual scenario but rather an isolated text alluding to green consumption. Additionally, it does not specify the situation in which it was produced, merely stating that it is "a post shared by someone on social media" (L339). This limitation is further exacerbated in the theoretical background (L137), where the role of influencers using such language and its subsequent impact on consumers’ perceptions is discussed without sufficient grounding.
Given this, it is evident that in a bragging discourse framed within a consumption context, the role of the post’s author is crucial in shaping individuals’ perceptions. This is particularly significant from a linguistic and pragmatic perspective, where the meaning and intention of discourse depend on both the producer and the receiver. Moreover, from a marketing standpoint, the study lacks experimental realism, which requires the replication of real-world conditions as closely as possible (van Heerde et al., 2021). Failing to properly execute this initial step in the experimental process may compromise the external validity of the results (Viglia et al., 2021).
Another concern pertains to participant eligibility. Their profiles do not indicate prior engagement with green consumption or any related behaviors. This lack of suitability is presented as part of the theoretical implications (L575-L578) rather than being acknowledged as a study limitation.
Author Response
Response to Reviewer 1
Thank you for your constructive feedback. We have carefully considered each of your suggestions and made revisions accordingly. Below are detailed responses to each of your comments.
- Comment: The concept of "Versailles literature language" lacks theoretical support in this study. The cited references do not include relevant studies that substantiate the application of this concept in the marketing field. Furthermore, the grouping of diverse literary figures under this concept appears deliberate and may only coincidentally align with the features used to define it. Therefore, it is important to note that there are insufficient metalinguistic elements to support the bragging intention of the post's author. Consequently, multiple interpretations can arise from the text, introducing potential bias based on the questions posed to participants.
Response:
Thank you for pointing out the deficiencies in the theoretical basis of this research, especially regarding the definition of the concept of “Versailles literaturel anguage style”. In response to this opinion, in this article, we have conducted a more systematic and detailed study on the origin, definition of the language style of "Versailles literature" and its theoretical support in related research (please refer to page 3, L103-L32, L142-L178).
In the revised version, we have added the definition of the language style of “Versailles literature” from a multidisciplinary perspective, including existing studies at multiple levels such as linguistics, psychology and sociology. And based on the research of previous scholars, this paper gives the definition of “Versailles literature” in the context of green consumption. Highlighting its high degree of consistency with the concept of “false modesty (humblebragging)” and comparing it to “direct boasting” and “ virtue signals” (e.g. Wittels, 2012; Grant et al., 2018; Berman et al., 2015). Furthermore, we have added the research on “Versailles literature ”in the field of marketing in the article to support its relevance in the fields of marketing and consumer perception. In the main text, we supplemented and explained the main characteristics of the “Versailles language” style, and emphasized that the recognition of this language style in specific contexts depends on semantic structure and expression methods rather than subjective speculation. To alleviate concerns about the possible introduction of bias in the identification of bragging intentions, we conducted a pre-experiment stage in the screening of experimental materials to verify the consistency of language style recognition through independent samples, in order to enhance the reliability and validity of the research (for relevant methods, please refer to L377-L382 on page 8).
- Comment: The primary issue lies in the construction of the stimuli. The post presented to participants does not provide any contextual scenario but rather an isolated text alluding to green consumption. Additionally, it does not specify the situation in which it was produced, merely stating that it is "a post shared by someone on social media" (L339). This limitation is further exacerbated in the theoretical background (L137), where the role of influencers using such language and its subsequent impact on consumers’ perceptions is discussed without sufficient grounding.
Response:
Thank you for pointing out the deficiencies in the background scenarios and information provided regarding the stimulating materials in our experimental design. We did have deficiencies in previous experiments. We have realized this and made the following improvements in subsequent Experiment Three based on your suggestions. In Experiment Three, we modified the stimulus materials and added detailed background descriptions. We added brief situational background explanations and clarified the Posting platform, the identity of the poster, and the reading environment. We also optimized the post texts used in the experiment to make them more in line with the language style of real social media platforms, and incorporated common social media elements (such as topic tags, etc.) into the content to enhance authenticity. Please refer to L607-L640 on page 15.
- Comment: Given this, it is evident that in a bragging discourse framed within a consumption context, the role of the post’s author is crucial in shaping individuals’ perceptions. This is particularly significant from a linguistic and pragmatic perspective, where the meaning and intention of discourse depend on both the producer and the receiver. Moreover, from a marketing standpoint, the study lacks experimental realism, which requires the replication of real-world conditions as closely as possible (van Heerde et al., 2021). Failing to properly execute this initial step in the experimental process may compromise the external validity of the results (Viglia et al., 2021).
Response:
Thank you for your in-depth criticism of this research. We fully agree with the reviewer's point that in the consumption context of social media discourse, the identity of the post author and the context setting are crucial for understanding the discourse intention and the audience's cognition. In the early experiments, there were indeed deficiencies in our setting of the author's role. Regarding this issue, we made modifications and optimizations in the subsequent experiment 3. In the stimulus materials, we clearly defined the identity of the post author and the social media platform where the post was posted to ensure that participants could clearly perceive the author's identity when reading the stimulus materials and enhance the authenticity and contextual fit of cognitive processing. Refer to the suggestions on the ecological validity of the experiment by van Heerde et al. (2021), and add preheating tasks or real social media browsing tasks before the experiment. We also enhanced the fidelity of the situation by adopting the interface styles, language styles and interaction contexts (such as the number of likes, comments, avatars and other elements) of real social platforms (Weibo and wechat), making the experiment closer to the usage environment of daily social media. During the experiment, we also measured the identity hierarchy of the poster and the recipient, and took them as control variables in the subsequent data analysis. Please refer to page 15 L607, page 16 L659.
- Comment: Another concern pertains to participant eligibility. Their profiles do not indicate prior engagement with green consumption or any related behaviors. This lack of suitability is presented as part of the theoretical implications (L575-L578) rather than being acknowledged as a study limitation.
Response:
Thank you to the reviewers for your valuable comments on the participants' green consumption experiences. In Experiment Three, we measured the green consumption behavior of the participants, specifically through questionnaires to determine their previous green consumption frequencies. This measurement result has been included as a control variable during the data analysis to ensure the robustness of the research conclusion. Please refer to page 15 L607 - page 16 L659.

Reviewer 2 Report
Comments and Suggestions for Authors
This manuscript investigated the influence of language style on consumer attitudes towards evaluating green consumption behavior and showed that the Versailles Literature language style reduces positive attitudes and increases perceptions of bragging and hypocrisy. While the topic is interesting and also significant for business owners in managing social media accounts, the quality of the manuscript does not meet the requirement of Behavioral Sciences.
- The introduction section is too lengthy. The authors are supposed to emphasize the significance and provide an overview of the manuscript concisely.
- While section 2 is “Theoretical Background”, the authors failed to critically discuss relevant theories and review the theory(ies) to be explored in the current study.
- For Section 3, the authors are recommended to provide further arguments for the proposed model. Based on previous research, why is the proposed model valid?
- More details should be provided regarding the experiments.
1) For Section 4.1.1, it is unclear what type of responses was collected. Was it just rating? Then how was it designed? Since the same materials were used in the formal experiment, there’s no point restating the two posts on Lines 341-352.
2) For Section 4.2, it is not mentioned how the factor of Social ties was manipulated.
- The design of the results lacks support from previous research. Also, the sample size is too small for questionnaire surveys.
- Another serious issue is that it is not recommended to use multiple t-tests, which will make the results unreliable. The authors should consider other types of statistical tests, such as regression, for the data analysis.
- Discuss of the results with previous studies is lacking. Section 6 did not relate to previous research.
Minor issues:
- The authors need to proofread the manuscript carefully, e.g. a direct quote from Line 157:
“evaluations(Effron et al., 2020)Error! Reference source not found.. In”
- Throughout the manuscript, a space is needed before the citation for each reference. Starting from Page 2, there’s no space at all.
- Some highly relevant key literature is missing. For example:
Ren, W., & Guo, Y. (2021). What is “Versailles Literature”?: Humblebrags on Chinese social networking sites. Journal of Pragmatics, 184, 185-195.
- Please add captions for Figures 2 to 4. The captions for table should also be more specified.
- Line 79: Please add citation to the self-presentation theory.
- Table 1: the letters in the box for Language Style cannot be seen completely.
- The text in the figures are hardly visible.
- “Pilot study/experiment” is a better term for “preliminary experiment”.
Author Response
Response to Reviewer 2
Thank you for your valuable comments. We have made significant changes to address each of your concerns and enhance the quality of the manuscript.
- Comment: The introduction section is too lengthy. The authors are supposed to emphasize the significance and provide an overview of the manuscript concisely.
Response:
Thank you very much to the reviewers for the structure of the introduction of this research. We have deleted and reconstructed the introduction, highlighting the importance and practical significance of the research. At the end of the paragraph, we have clearly summarized the structure of the article and the main research content. Please refer to page 1- page 2 L89.
- Comment: While section 2 is “Theoretical Background”, the authors failed to critically discuss relevant theories and review the theory(ies) to be explored in the current study.
Response:
Thank you to the reviewers for your detailed review and constructive suggestions on the "Theoretical Background" section of Section 2. We have made significant revisions to this part based on your opinions. In the revised version, we have added the definition of the language style of Versailles literature from a multidisciplinary perspective, including existing studies at multiple levels such as linguistics, psychology and sociology, and given the definition of Versailles literature in the context of green consumption in this paper based on the research of previous scholars. Highlighting its high degree of consistency with the concept of "false modesty (humblebragging)" and comparing it to "direct boasting" and "virtue signals" (e.g. Wittels, 2012; Grant et al., 2018; Berman et al., 2015). Furthermore, we further traced the theoretical source of this language style (Goffman's "self-presentation" theory), and we also systematically reviewed the research progress of this language style in consumer behaviors such as brand attitude and purchase intention. Please refer to page 2 L92- page 4 L178.
- Comment:For Section 3, the authors are recommended to provide further arguments for the proposed model. Based on previous research, why is the proposed model valid?
Response:
Thank you to the reviewers for your attention to the model construction part of this research. We understand that the explanation of the theoretical basis for model construction in the original manuscript is still insufficient and fails to fully clarify the theoretical and empirical basis of the relationship among various variables. Based on your suggestions, we systematically supplemented and strengthened each path relationship in the model one by one in Section 3 of the revised draft. We further cite research on self and the "pseudo-modesty" (humblebragging) language style, which is prone to raise questions about the speaker's motives and hypocritical perceptions among the audience, which in turn affects the acceptance of the information it conveys. Combining moral psychology with social comparative research, we illustrate that hypocritical perception may weaken the audience's attitude towards green behavior or brands by undermining the speaker's moral credibility. This perception of showing off can trigger consumers' social comparison psychology, making them feel belittled or unequal, and thus generating a negative attitude towards the poster's green consumption behavior. Please refer to page 5 L235- page 6 L275
- Comment:More details should be provided regarding the experiments.
(1) For Section 4.1.1, it is unclear what type of responses was collected. Was it just rating? Then how was it designed? Since the same materials were used in the formal experiment, there’s no point restating the two posts on Lines 341-352.
(2)For Section 4.2, it is not mentioned how the factor of Social ties was manipulated.
Response:
(1)Thank you to the reviewers for pointing out the problem that our description of the predictive experimental design in Section 4.1.1 is not clear enough. In response to the questions you raised, we have supplemented the measurement of the language style of Versailles literature that was already in the revised draft, and added the participants' views and ratings of Versailles literature in Experiment Three. We also deleted the same materials in Experiment 1 as those in the Pilot experiment. Please refer to page 8 L377-L382.
(2) Thank the reviewers for pointing out that Section 4.2 lacks a discussion on the longitudinal influence of social relationship factors. In our previous experiments, we directly manipulated the degree of familiarity through guiding words in the situational materials. According to your opinion, we have made optimizations in Experiment Three. In Experiment 3, we adopted the characteristics of real social platforms themselves (wechat vs. Weibo) as the manipulation medium and conducted measurements using questionnaires.“I think wechat/Weibo is a kind of strong relationship social platform. The wechat/Weibo platform is more about people I am familiar with.” Please refer to page 16 L643-L659.
- Comment:The design of the results lacks support from previous research. Also, the sample size is too small for questionnaire surveys.
Response:
Thank you very much for your valuable suggestions. We have included more previous studies related to the design of this research in the revised draft to support us. We have supplemented the literature review on green consumption behavior, the language style of social media, and the language style of "Versailles literature" to ensure that the research methods and design have a solid theoretical and empirical basis.
The Cohen's d value obtained by the independent sample t-test represents the size of the effect and indicates the magnitude of the difference between groups. The larger the cohen's d value is, the greater the difference indicates. In this study, Cohen's d values all exceeded 0.8, indicating that the effect size was large and there was sufficient statistical power. We also conducted the analysis using GPower. The analysis showed that when performing the independent sample t-test, the statistical power of 60 samples was 0.85, the significance level was 0.05, and the effect size was 0.80, indicating that this study has sufficient power and the sample size is adequate. When conducting the analysis of variance, the statistical power of 120 samples was 0.90, the significance level was 0.05, the effect size was 0.30, and 0.25 was considered a moderate effect when conducting the analysis of variance. Therefore, this study has sufficient efficacy and the sample size is adequate.
- Comment:Another serious issue is that it is not recommended to use multiple t-tests, which will make the results unreliable. The authors should consider other types of statistical tests, such as regression, for the data analysis.
Response:
Thank you very much for your suggestions. We have supplemented the data analysis results. We have supplemented the demographic characteristics and regression analysis results.
- Comment:Discuss of the results with previous studies is lacking. Section 6 did not relate to previous research.
Response:
Thank you to the reviewers for your valuable suggestions in the conclusion section. We clearly pointed out in the conclusion that this study found that the language style of "Versailles literature" significantly enhanced consumers' perception of showing off and hypocrisy, thereby reducing their attitude towards the green consumption behavior of the poster. This is largely consistent with the research of Berger & Ward (2010) on the impact of conspicuous consumption behavior. Yan etal. (2016) emphasized that the closeness of social relationships, as an important dimension of psychological distance, can affect the way the audience interprets the motivation of information. The research found that consumers are more likely to perceive the Versailles-style content posted by strangers as showing off and hypocrisy when faced with it. This finding is highly consistent with the existing theoretical framework. Please refer to page 19 L739- page 20 L762.
- Comment: (1)The authors need to proofread the manuscript carefully, e.g. a direct quote from Line 157:“evaluations(Effron et al., 2020)Error! Reference source not found.. In”
(2) Throughout the manuscript, a space is needed before the citation for each reference. Starting from Page 2, there’s no space at all.
(3) Some highly relevant key literature is missing. For example:Ren, W., & Guo, Y. (2021). What is “Versailles Literature”?: Humblebrags on Chinese social networking sites. Journal of Pragmatics, 184, 185-195.
(4)Please add captions for Figures 2 to 4. The captions for table should also be more specified.
(5)ine 79: Please add citation to the self-presentation theory.
(6)Table 1: the letters in the box for Language Style cannot be seen completely.
(7)The text in the figures are hardly visible.
(8) “Pilot study/experiment" is a better term for "preliminary experiment".
Response:
Thank you very much for your suggestions. We made revisions based on your suggestions. Firstly, for the error in line 157, we modified the statement and replaced the literature again. We have also added the relevant literature you recommended and the citations of the self-presentation theory. We have also re-explained the title of the table. In addition, for the issue where the text in the picture is not clear, we have presented it in the form of a table. Finally, we have changed "preliminary experiment" in the text to "Pilot experiment".

Reviewer 3 Report
Comments and Suggestions for Authors
This study explored the impact of language style (Versailles literature vs. non-Versailles literature) and the strength of social ties on consumers' attitudes toward green consumption behavior, examining the mediating roles of perceptions of bragging and hypocrisy. However, there are several points of concern:
1. The definition of the specific language style referred to as "Versailles literature" and its significance is insufficiently explained, which may lead to difficulties for readers in understanding this concept. A more detailed theoretical framework is necessary.
2. The explanation of the hypotheses is overly simplistic and does not adequately reflect the background of the study. It is essential to more specifically incorporate the research context into the hypotheses.
3. There is a lack of validation regarding whether the experimental stimuli accurately reflect the Versailles language style. An analysis is needed to determine if the texts used in the preliminary experiment clearly differentiate between the two language styles. Additionally, a method for assessing how the Versailles style is perceived by consumers should be established. It is particularly unclear what variables were considered when analyzing the differences between the two conditions in the manipulation check.
4. The conclusion should focus on deriving implications by comparing the results of this study with those of prior research. Furthermore, references to AI chat or related technologies are not closely related to the core topic of this study. It would be advisable to remove this section to maintain the focus of the research on directly relevant content.
I think addressing these points will strengthen the clarity and impact of the study. Good luck with your research!
Author Response
Response to Reviewer 3
Thank you for your detailed review of our manuscript and valuable suggestions. The opinions you put forward are of great significance for us to further improve the research. We have made corresponding revisions and supplements to the manuscript based on your feedback. The responses are as follows one by one:
- Comment:The definition of the specific language style referred to as "Versailles literature" and its significance is insufficiently explained, which may lead to difficulties for readers in understanding this concept. A more detailed theoretical framework is necessary.
Response:
Thank you to the reviewers for your valuable comments on our research. In the revised version, we have added the definition of the language style of Versailles literature from a multidisciplinary perspective, including existing studies at multiple levels such as linguistics, psychology and sociology, and given the definition of Versailles literature in the context of green consumption in this paper based on the research of previous scholars. Highlighting its high degree of consistency with the concept of "false modesty (humblebragging)" and comparing it to "direct boasting" and "virtue signals" (e.g. Wittels, 2012; Grant et al., 2018; Berman et al., 2015). Please refer to page 3, L103-L132.
- Comment:The explanation of the hypotheses is overly simplistic and does not adequately reflect the background of the study. It is essential to more specifically incorporate the research context into the hypotheses.
Response:
Thank you to the reviewers for your valuable comments on the hypothesis part of our research. In the revised draft, we have provided a more detailed explanation of the background of each hypothesis and conducted sufficient arguments in combination with relevant literature and theories. We further cite research on self and the "pseudo-modesty" (humblebragging) language style, which is prone to raise questions about the speaker's motives and hypocritical perceptions among the audience, which in turn affects the acceptance of the information it conveys. Combining moral psychology with social comparative research, we illustrate that hypocritical perception may weaken the audience's attitude towards green behavior or brands by undermining the speaker's moral credibility. This perception of showing off can trigger consumers' social comparison psychology, making them feel belittled or unequal, and thus generating a negative attitude towards the poster's green consumption behavior. Please refer to page 5 L235- page 6 L275
- Comment:There is a lack of validation regarding whether the experimental stimuli accurately reflect the Versailles language style. An analysis is needed to determine if the texts used in the preliminary experiment clearly differentiate between the two language styles. Additionally, a method for assessing how the Versailles style is perceived by consumers should be established. It is particularly unclear what variables were considered when analyzing the differences between the two conditions in the manipulation check.
Response:
Thank the reviewers for pointing out our problems. In response to the questions you raised, we conducted a pre-experiment before the experiment and supplemented the measurement of the language style of Versailles literature in the revised draft. In Experiment Three, we added the participants' views and ratings of Versailles literature. “Do you think the description of the content of this post is natural and simple or exag-gerated and showy? " “Do you think this text represents a “Versailles Litera-ture”language style? (“Versailles Literature”language style refers to the use of apparent modesty, complaint, or self-deprecation to subtly convey one’s superior status or achievements).” Please refer to page 8 L377-L382.
Furthermore, we had already included the control variables when conducting the data analysis before, but we omitted this part in the text. Regarding this, we have explained the control variables for the data analysis in the experiment in the text.
- Comment:The conclusion should focus on deriving implications by comparing the results of this study with those of prior research. Furthermore, references to AI chat or related technologies are not closely related to the core topic of this study. It would be advisable to remove this section to maintain the focus of the research on directly relevant content.
Response:
Thank you very much for your suggestions on the conclusion section. We clearly pointed out in the conclusion that this study found that the language style of "Versailles literature" significantly enhanced consumers' perception of showing off and hypocrisy, thereby reducing their attitude towards the green consumption behavior of the poster. This is largely consistent with the research of Berger & Ward (2010) on the impact of conspicuous consumption behavior. Yan etal. (2016) emphasized that the closeness of social relationships, as an important dimension of psychological distance, can affect the way the audience interprets the motivation of information. The research found that consumers are more likely to perceive the Versailles-style content posted by strangers as showing off and hypocrisy when faced with it. This finding is highly consistent with the existing theoretical framework. I have deleted the irrelevant AI content you mentioned in the text. Please refer to page 19 L739- page 20 L762.

Reviewer 4 Report
Comments and Suggestions for Authors
General Comment
Overall, this article is a valuable and timely contribution to the study of digital communication, green consumption, and consumer perception. It introduces an innovative concept in "Versailles Literature" and examines it with theoretical depth and empirical rigor. There are areas where the writing could be more concise and the presentation more polished, particularly in the figures and some theoretical explanations. Nonetheless, the study offers both theoretical advancement and practical utility. With minor revisions for clarity and structure, this paper is well-suited for publication and likely to stimulate further research in this emerging area.
Below are comments for each section:
Abstract
The abstract provides a coherent overview of the study’s rationale, methodology, and findings. It effectively communicates that the paper focuses on how language style, particularly a boastful or humblebrag style known as Versailles Literature, affects consumer perceptions of green consumption shared on social media. However, the abstract would benefit from briefly defining Versailles Literature, as this term may not be familiar to all readers, especially those outside Chinese digital culture.
Additionally, the abstract uses technical terms like “greenwashing” without clarifying them, which could alienate readers unfamiliar with the jargon. A sentence or two to clarify the meaning or context of these terms would enhance accessibility.That’s a suggestion.
Introduction
The introduction provides a strong foundation by clearly situating the study within current global and technological trends. It effectively explains why green consumption is socially important and why social media plays a significant role in promoting it. The narrative transitions naturally from broader concerns (like environmental degradation and online communication) to the specific research focus. The use of examples is helpful and engaging.
That said, the section tends to reiterate the same point about exaggerated green behaviors in slightly different ways, which creates some redundancy. It could be more concise by trimming repetitive sentences and sharpening the articulation of the research gap. The literature cited is relevant, but the integration could be more structured. For instance, rather than weaving related studies into multiple paragraphs, summarizing the major findings of past work and then identifying the gap in a focused paragraph would improve clarity.
Theoretical Background
The section on language style and Versailles Literature offers a thoughtful conceptualization of language’s social functions in digital settings. The explanation of Versailles Literature is informative, pointing to its ironic, humblebragging tone and how it conveys superiority under the guise of humility. This is well-illustrated through references to social media posts, making the theory accessible. The theoretical linkage to self-presentation theory is appropriate and helps contextualize the motivations behind this linguistic style.
However, the description of the literature could be further enriched by comparing Versailles Literature to other known styles, such as direct boasting or virtue signaling, to highlight its uniqueness more effectively.
The discussions on hypocrisy perception and bragging perception are grounded in existing literature and well-aligned with the study's aims. These subsections are logically organized and offer useful definitions, but they sometimes repeat concepts that were already touched upon earlier in the article.
Streamlining the theory section to reduce overlap and more clearly differentiate between these psychological constructs would help sharpen the theoretical focus.
Research Hypotheses and Conceptual Model
Very good job here. The only comment I have is the justification of moderators and the fact that some of the hypotheses are lengthy and could be stated more succinctly to improve readability.
Methodology
More detail on sample demographics and recruitment would enhance transparency. Although the authors note the use of university students, it would help to include a table summarizing participant characteristics to provide a clearer view of the sample.
Results
The results section is comprehensive and statistically robust. The authors use a range of appropriate statistical tests and present the findings clearly. Both direct and indirect effects are discussed, and manipulation checks are well implemented, providing confidence in the integrity of the experimental manipulations.
Breaking up the section with clearer subheadings or brief summary sentences after each statistical test could enhance clarity.
Conclusion and Discussion
The conclusion section succinctly recaps the main findings and appropriately links them back to the study’s hypotheses. The authors do an excellent job of showing how their findings contribute to a broader understanding of consumer psychology and social media behavior. However, some conclusions are repeated from earlier sections, which reduces the overall punch of the discussion. A more concise summary focusing on unique contributions and implications would be more impactful.
Theoretical Contributions and Managerial Implications
This section provides rich insights into the interdisciplinary implications of the findings. The theoretical contributions are well-articulated, especially the blending of social psychology with green consumer behavior. The managerial implications are practical and well-tailored to digital marketing professionals. No further comments for improvement.
Limitations and Future Research
The authors are transparent about the study’s limitations, including sample representativeness, limited language styles, and platform-specific behaviors. The proposed directions for future research are forward-thinking and relevant. However, some ideas, like the integration of AI and chatbot-generated language, feel speculative and less directly tied to the current study’s scope. Further linking these suggestions back to the core variables of language style, self-presentation, and perception would make them more coherent.